# Cellular and genetic mechanisms that shape the development and evolution of tail vertebral proportion in mice and jerboas

Ceri J. Weber [1], Alexander J. Weitzel [1], Alexander Y. Liu [1], Erica G. Gacasan[2], Susan C. Chapman [3], Robert L. Sah [2] & Kimberly L. Cooper [1]

Limbs and vertebrae elongate by endochondral ossification, but local growth control is highly modular such that not all bones are the same length. Compared to limbs, which have a different evolutionary and developmental origin, far less is known about how individual vertebrae establish proportion. Using the jerboa and mouse tail skeletons, we find that cell number is a common driver of limb and vertebral proportion in both species. However, chondrocyte hypertrophy, which is a major driver of proportion in all mammal limbs, is limited to the extreme disproportionate growth of jerboa mid-tail vertebrae. The genes associated with differential growth in the vertebral skeleton overlap significantly, but not substantially, with genes associated with limb proportion. Among shared candidates, loss of Natriuretic Peptide Receptor 3 in mice causes disproportionate elongation of the proximal and mid-tail vertebrae, in addition to the proximal limb. Our findings therefore, reveal cellular processes that tune the growth of individual vertebrae while also identifying natriuretic peptide signaling among genetic control mechanisms that shape the entire skeleton.

Mammal skeletal diversity is remarkable. Much attention is paid to differences in the limbs and skull bones, but the vertebral skeleton is also strikingly different between species. For example, humans, dolphins, and giraffes all have seven cervical vertebrae, but very different neck lengths[1,2]. In buffalo, extremely elongated neural spines extend from the dorsal side of the thoracic vertebrae and serve as muscle attachment sites to support its massive head. At the far opposite end of the axial skeleton, tails range from externally absent in hominoid primates to elongate and prehensile appendages in New World monkeys[3]. How do the differences between vertebral size and shape develop and evolve?

Different parts of the vertebrate skeleton have distinct embryonic origins. The skull derives from neural crest and cephalic and somitic mesoderm[4], the limbs emerge from lateral plate-derived buds[5], and the vertebral/axial skeleton develops from the embryonic somites[6,7]. A substantial body of research has revealed the mechanisms of somitogenesis, which pinches off blocks of tissue in the trunk and tail region. A molecular clock determines the cadence of somite formation[8–11], and maintenance of a progenitor pool determines the number of somites and thus the number of vertebrae, forming the extraordinarily long vertebral column of snakes in extreme cases[12,13].

As the metameric series of somites marches down the body axis, they acquire regional identities by translating their anterior-posterior position into expression of a "Hox code"[14–16]. Hox genes are anatomically expressed in reverse collinear order of their appearance in the genome, with 3' Hox genes in the anterior somites and 5' Hox genes appearing posteriorly. These genes are both necessary and sufficient to define regional identities for groups of cervical (neck), thoracic (rib

[1]Department of Cell and Developmental Biology, University of California San Diego, La Jolla, CA, USA. [2]Shu Chien-Gene Lay Department of Bioengineering, University of California San Diego, La Jolla, CA, USA. [3]Department of Biological Sciences, Clemson University, Clemson, SC, USA. ✉e-mail: cweber@ucsd.edu; kcooper@ucsd.edu

cage), lumbar (lower back), sacral (pelvic), and caudal (tail) vertebrae. For example, loss of function of the entire *Hox10* paralogous group results in lumbar vertebrae taking on a thoracic morphology with rib processes projecting from each element[17].

However, once somites transform into cartilaginous vertebral scaffolds that will later become ossified bone, little is known about how they acquire distinct shapes and sizes. Individual vertebrae comprise a centrum with transverse, dorsal (neural), and/or ventral (hemal) processes[7]. The processes form articulations between neighboring vertebrae and between vertebrae and ribs, muscle attachment sites, and protective structures for the spinal cord or dorsal artery, giving distinctive shapes to each vertebra. Despite the relatively simple geometry of the centrum, the cylindrical core that lines up in series down the axis, its length differs substantially between axial regions and even between neighboring vertebrae[18].

Additionally, the ossified vertebral skeleton pre-dates the limb skeleton in its evolutionary origin by at least 60 million years[19–21]. Despite these differences, elongation of both limb bones and vertebrae occurs by a process of endochondral ossification of growth cartilages[22]. However, while advances have been made to understand the cellular and genetic mechanisms that drive local differences in limb bone elongation, far less is known of the mechanisms that control differential growth of the vertebral centra. How similar or different are the cellular and genetic mechanisms that control proportion in the limb and vertebral skeleton?

To answer these questions, we use the bipedal lesser Egyptian jerboa (*Jaculus jaculus*) as a model of extreme musculoskeletal proportion compared to quadrupedal laboratory mice. Thus far, much of our attention has focused on the extreme elongation of the hindlimb, particularly the disproportionately long feet. However, the jerboa also has a disproportionately long tail, approximately 1.5-times longer than the mouse, normalized to body length[23]. Surprisingly, the long jerboa tail has three to four fewer vertebrae than in mice; their long tails are acquired by far greater elongation of individual vertebral elements in the mid-tail region. This substantially exaggerates the "crescendo-decrescendo" of vertebral proportions reported in the tail series of rodents, primates, and carnivores[3,18,24–27].

Here, we use the jerboa and mouse to understand the temporal growth dynamics that establish adult vertebral proportion, the cellular drivers of differential growth, and candidate genetic mechanisms that determine and diversify vertebral proportion. We find that a greater number of chondrocytes undergoing endochondral ossification primarily drives differential elongation of the centrum, which is amplified by greater chondrocyte hypertrophy only in the case of extremely long jerboa vertebrae. Intersectional differential expression analyses revealed that most genes disproportionately expressed in vertebrae are not also found in studies of limb bone proportion. Still, there is statistically significant overlap, suggesting a subset of gene regulatory networks control proportion throughout the skeleton.

Among the shared candidate genes in these networks, natriuretic peptide receptor C (*Npr3*) appears in similar studies of jerboa, mouse, and rat limb proportion and a GWAS analysis of human body proportion[28–31]. We show that loss of NPR3 in the mouse causes disproportionate elongation of the proximal tail vertebrae through expansion of the hypertrophic chondrocyte zone and increased hypertrophic chondrocyte cell size. These data suggest that this pathway may be a crucial regulator of proportion in the vertebral skeleton, in addition to its previously implicated role in the development of limb proportion, through regulation of chondrocyte differentiation. Building on previously observed differences between the tail skeletons of adult mice and jerboas, our work has identified cellular and genetic mechanisms that may establish vertebral proportion and lays a foundation to uncover additional mechanisms of skeletal modularity that shape the extraordinary diversity of mammals.

## Results

### Development of adult tail proportion

We previously reported that the adult jerboa tail is 1.5-times the length of a mouse tail, normalized to body length, with three to four fewer vertebrae[23]; vertebral counts vary by one in both species (Fig. 1A, B). To achieve this, the vertebrae in the jerboa mid-tail region reach far greater lengths, disproportionately increasing the overall tail length relative to body length. To identify when differences in overall tail length and vertebral proportion within the tail first appear and how they manifest, we collected mice and jerboas at weekly intervals from birth to mature tail proportion at six weeks of age (postnatal day 42; P42). We measured the ratio of overall tail length to body length (naso-anal distance) and used micro-computed tomography (μCT) images to measure the lengths of each ossified tail vertebral centrum (diaphysis) in the anterior-posterior axis (Fig. 1).

At birth, the normalized lengths of mouse and jerboa tails are both approximately half of the naso-anal distance (Figs. 1C, D and S1), and individual vertebrae are sequentially shorter from the first tail vertebra to the tip of the tail in both species (Fig. 1E, F). Interestingly, although both species have formed all the cartilaginous vertebral scaffolds by birth, 31 in mice and 28 in jerboas ($n = 2$ each), only half as many tail vertebrae are ossified in jerboas compared to mice (Fig. 1C, E, F). A similar delay in ossification has been observed associated with extreme elongation of jerboa metatarsals and bat metacarpals[32–34].

These data suggest that the difference in total tail length and vertebral proportion, within and between species, does not emerge during somitogenesis or early chondrogenesis but rather during postnatal endochondral ossification of the centra. From birth to P14, the overall tail length to body length ratio maintains a similar proportion in the two species (Figs. 1E, F and S1). At P7 in both mice and jerboas, a disproportionately rapid rate of elongation is initiated in the mid-tail region plateauing from TV5 to TV8 with TV6 demonstrating the greatest change in vertebral length in each species (Fig. 1E–H). Vertebral lengths then sequentially decrease to the tail tip. This "crescendo-decrescendo" is amplified by continued disproportionate growth postnatally, far more rapidly in jerboas, and adult tail proportion is achieved in both species by P35 (Fig. 1E, F). In jerboas, the most rapid disproportionate elongation of mid-tail vertebrae occurs between P14 and P21, the time when the overall tail length to body ratio also diverges from that of the mouse (Fig. 1D, H).

Interestingly, this period of the most rapid vertebral elongation occurs one week later in jerboas than in mice, consistent with our observation that the initial ossification in the tail series and formation of the secondary ossification centers (vertebral endplates) are also delayed in jerboas compared to mice (Fig. S2). This supports the hypothesis that disproportionately rapidly elongating skeletal elements in jerboa maintain a younger growth cartilage, enabling faster growth for longer than the homologous elements of mice[32].

### Cellular parameters of tail vertebral elongation

Each vertebral centrum elongates by endochondral ossification of the cranial and caudal growth cartilages, though not necessarily at the same symmetric rate. To quantify the relative rates of growth contributing to vertebral elongation, we focused on the vertebrae with the most similar (tail vertebrate 1; TV1) and most different rates of elongation (TV6) between species in the proximal to mid-tail (Fig. 2A–C). We selected a timepoint within the window of greatest disproportionate elongation for each species before formation of the secondary ossification centers, P5–7 for mouse and P14–16 for jerboa (Fig. S2). To quantify the daily rate of growth cartilage elongation during this growth phase, we injected pups with a pulse of calcein dye to fluorescently label mineralized bone (at P5 for mouse and P14 for jerboa), two days before collection at P7 or P16 (Fig. S3). The distance between the calcein-labeled bone and the chondro-osseous junction of the

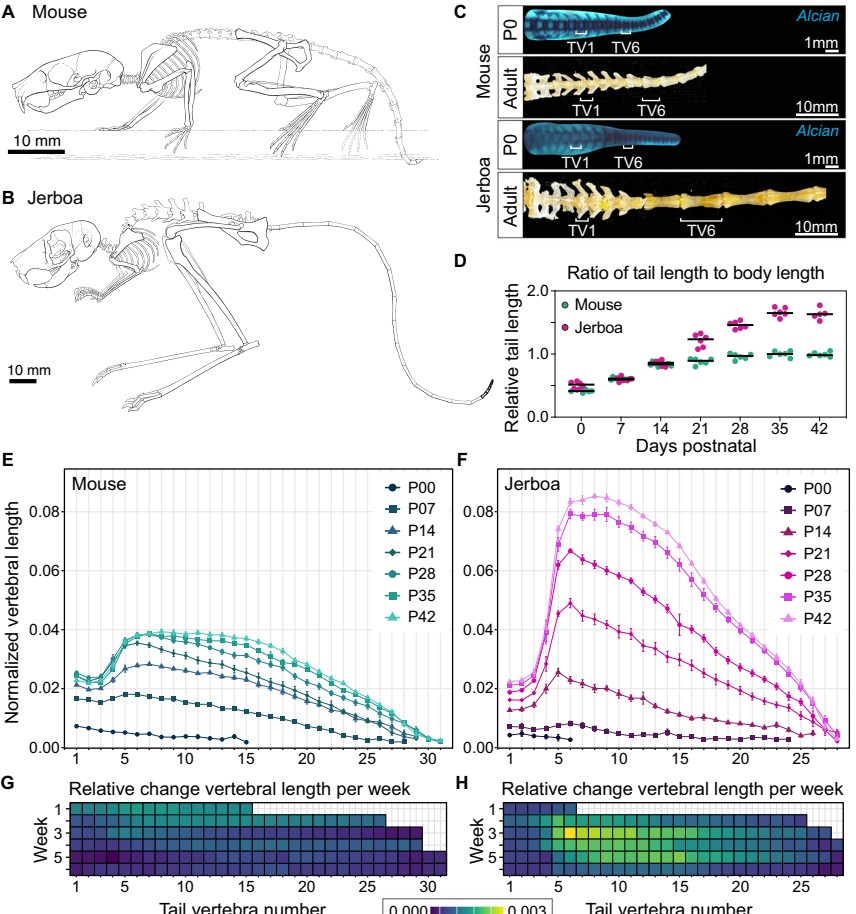

**Fig. 1 | Development of mouse and jerboa tail proportion. A**, **B** Diagram of adult mouse (**A**) and jerboa (**B**) skeletons adapted from Moore et al. 2015. **C** Alcian-stained neonates show all proximal vertebral elements are present at birth (P0) in both species. Adult proximal tail skeletons have similar vertebral morphologies despite differences in size and proportion. **D** Mouse (teal) and jerboa (pink) tails are approximately half the naso-anal length at birth. Tail proportion diverges by P21; the mouse tail remains about equal to body length, while the jerboa tail elongates to 1.5-times the body length. **E**–**H** μCT scans were used to measure vertebral centra lengths weekly from birth to six weeks, normalized to the naso-anal length of each mouse (**E**) and jerboa (**F**). **E**, **F** Points represent the mean vertebral lengths at each position and timepoint with error bars showing the standard deviation of these measurements. Each timepoint from P0 to postnatal day 42 (P42) is represented by a different shape and color for mouse (blues) and jerboa (pinks). The weekly relative change in length of each vertebra, normalized to naso-anal length, is represented in a heat map. The greatest rate of change is yellow and least in dark blue with the scale equivalent for mouse (**G**) and jerboa (**H**). **D**–**H** Six male and female animals were measured for each time point, except jerboa P0 and P42 where *n* = 5. Source data are provided as a Source Data file.

growth cartilage, divided by two, reveals the growth rate as the length of new bone formed per 24-h period.

During this window of greatest differential growth of TV6 versus TV1, we found that cranial growth cartilage elongation is significantly slower than caudal elongation within mouse TV1. Conversely, the cranial growth cartilage elongates slightly but significantly faster than the caudal growth cartilage in jerboa TV1 (Figs. 2D and S3). There is no significant difference between the cranial and caudal growth rates within TV6 of either species. However, consistent with μCT measurements of whole vertebrae, both cranial and caudal growth cartilages elongate significantly faster in TV6 than in TV1 in both species, and the jerboa TV6 growth cartilages elongate more than two-times faster than the mouse.

In reptiles and birds, differential growth primarily correlates with the number of cells in the growth cartilage, particularly the height of the proliferative zone[35,36]. In mammal limbs, these differences are additionally amplified by differences in final hypertrophic cell size[33,34,37–41]. The doubling of an individual flattened proliferative chondrocyte adds just 8–9 μm to the axis of elongation, whereas each of these cells additionally adds up to 40–50 μm to the growth axis as it

proceeds through hypertrophic enlargement[33,38]. We therefore measured growth cartilage height, proliferative zone height and proliferation index, and hypertrophic zone height and maximum hypertrophic chondrocyte size to distinguish relative cellular contributions to differential vertebral elongation in cranial and caudal growth cartilages of TV1 and TV6 in mouse and jerboa.

We first measured total growth cartilage height from the intervertebral joint surface to the chondro-osseous junction. In both species, growth cartilage heights are far greater in the more rapidly elongating TV6 than in TV1 (Figs. 2B–E and S3). Consistent with the greater than twofold difference in growth rate, both cranial and caudal growth cartilages of jerboa TV6 are more than twice the height of any other measured growth cartilage. The heights of individual zones within the growth cartilage also coincide with growth rate (Fig. 2F, H).

A greater proliferative zone height could be due to faster cell-cycle time that produces more cells before they initiate hypertrophic differentiation, and/or a larger number of progenitor cells. To test this, we measured the proliferation index within each growth cartilage during the rapid growth window by injecting EdU into pups two hours before collection at P7 (mouse) or P16 (jerboa) to label cells in S-phase

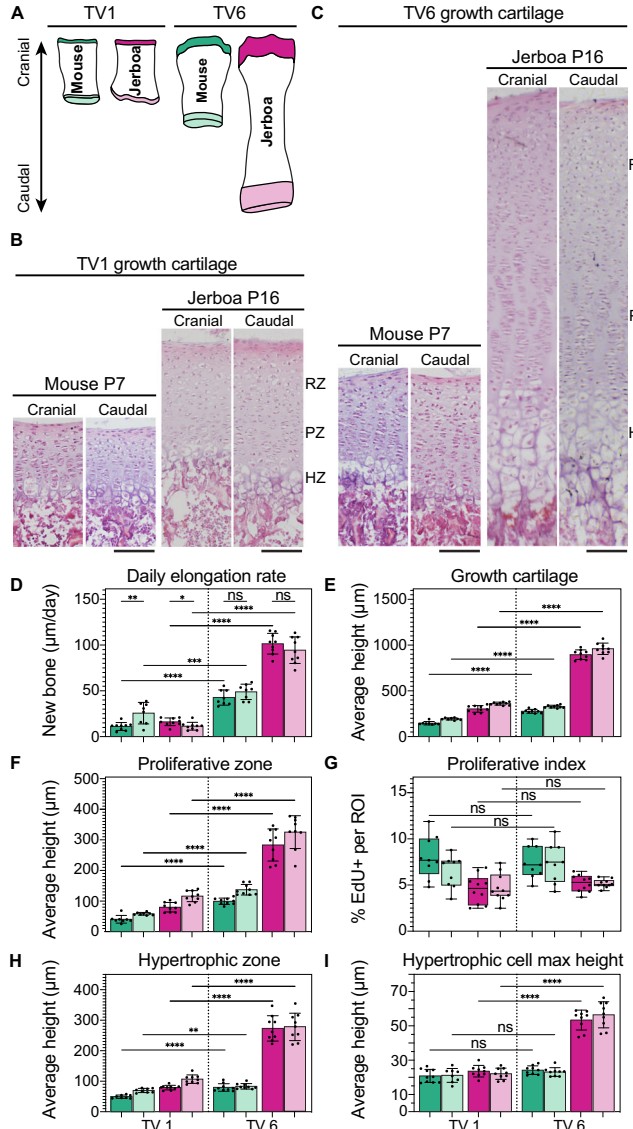

**Fig. 2 | Cellular parameters of growth during the greatest difference in vertebral elongation rate. A** Schematic of tail vertebrae with cranial and caudal growth cartilage color schemes used throughout. **B**, **C** Hematoxylin and eosin staining of mouse and jerboa cranial and caudal growth cartilages of tail vertebra 1 (TV1) (**B**) and TV6 (**C**) during rapid elongation. (RZ resting zone, PZ proliferative zone, HZ hypertrophic zone). Scale bar is 100 μm. **D**–**I** Growth cartilage parameters in each cranial and caudal cartilage of mouse and jerboa TV1 and TV6. **D** Daily elongation rate, $n = 8$ for mouse TV1 caudal and mouse TV6 cranial and caudal, $n = 9$ for mouse TV1 cranial and all jerboa TV1 and TV6 cartilages. **E** Height of the growth cartilage. **F** Height of the proliferative zone. **G** Proliferative index calculated as the fraction of EdU+ cells of all cells in an ROI. For mouse cartilages $n = 9$, and $n = 8$ for all jerboa cartilages. **H** Height of the hypertrophic zone. **I** Maximum height of hypertrophic chondrocytes in the direction of bone elongation. **D**–**I** Welch's *t*-test, two-tailed, * ≤ 0.05, ** ≤ 0.01, *** ≤ 0.001, **** ≤ 0.0001. Exact *p*-values are available in Source Data file. **E**, **F**, **H**, **I** $n = 8$ for mouse TV1 cranial and caudal cartilages, $n = 9$ for mouse TV6 and all jerboa TV1 and TV6 cartilages, all mixed male and female animals. **D**–**F**, **H**, **I** Error bar centers are the data means and error bands show standard deviation. **G** Box and whisker plot boxes extended from the 25th to 75th percentiles with whiskers showing the minimum and maximum values, the center line is the median, and all points are plotted. Source data are provided as a Source Data file.

of the cell cycle. A greater fraction of cells in S-phase would indicate a more rapid division rate. However, we found that the fraction of S-phase chondrocytes is not significantly different between TV6 and TV1 in either species (Figs. 2G and S4). This suggests that the difference

in proliferative zone height is due to a greater number of chondrocytes dividing at a similar rate rather than a faster rate of progression through the cell cycle.

Consistent with a hypothesis that more cells progress through endochondral ossification of the most rapidly elongating growth cartilages, we found that the height of the hypertrophic zone is also larger (Fig. 2H). To assess the contribution of hypertrophic chondrocyte size to differential growth of vertebrae, we measured the average maximum diameter of these cells in the axis of elongation in cranial and caudal growth cartilages of mouse and jerboa TV1 and TV6. Hypertrophic cell size is only marginally or not significantly different in mouse growth cartilages and in jerboa TV1. However, the extreme difference in jerboa TV6 elongation versus other vertebral growth cartilages is driven in part by hypertrophic chondrocytes that are more than twice the height of other vertebral chondrocytes (Fig. 2I). This was surprising, because even hypertrophic chondrocytes in the rapidly elongating jerboa metatarsus are only ~58% larger than those in the mouse, and indeed the difference in vertebrae is because slower elongating vertebrae have much smaller cells[33]. This suggests that hypertrophic chondrocyte size is evolvable in mammalian vertebral growth cartilages, but it might only drive extreme vertebral proportion, whereas it contributes to limb bone proportion broadly in mammals.

Collectively, these data suggest that the tail "crescendo-decrescendo" is largely driven by differences in the number of cells progressing through endochondral ossification in each vertebra, as in bird and reptile limb bones, and this difference is greatly amplified by larger hypertrophic chondrocytes in jerboa TV6 driving extreme disproportionate elongation of the mid-tail. We then sought to identify candidate genes in regulatory networks that may cause the cellular differences that drive the development and evolution of vertebral proportion.

## Interspecies intersectional transcriptomics reveal candidate genetic mechanisms of tail diversification

Multiple studies have investigated the genetic basis of skeletal proportion by identifying gene expression differences in limb bone growth cartilages[29,30,42]. Previously, we applied such an approach to identify candidate drivers of limb skeletal proportion in the jerboa, with its elongated hindlimb and disproportionately long feet compared to mice[30]. Far less is known about the genetic mechanisms of vertebral elongation to establish proportion or to what extent these mechanisms are shared with long bones of the limb. A similar careful intersection of comparative gene expression studies can identify candidate genetic networks and biological processes for the modular control of vertebral elongation to establish axial proportion.

Because the cranial and caudal contributions to vertebral elongation are very similar in all growth cartilages, but the difference in relative contribution to growth is greatest between cranial growth cartilages of mouse TV1 and TV6 (Fig. 2D), we chose to focus on cranial growth cartilages for these analyses. We performed bulk-RNA sequencing on three pooled cartilages per each of four replicates during the windows of greatest disproportionate growth in mouse and jerboa, at P7 and P16, respectively, before the appearance of secondary centers of ossification (Figs. 3A and S2). To directly quantify differences in gene expression between jerboa and mouse homologous skeletal elements, we first annotated a set of 17,640 orthologous genes using TOGA and the *Mus musculus* (mm10) and revised *Jaculus jaculus* (mJacJac1.mat.Y.cur) genome assemblies (NCBI)[43,44]. We included one-to-one orthologous transcripts and also genes in the one-to-zero classification, because sequence and/or assembly errors might explain predicted functional loss.

For each of the four sample sets, mouse and jerboa TV1 and TV6, we sequenced and mapped reads to the respective genome using the TOGA orthologous gene set annotations. We then used DESeq2 to perform differential expression analyses within and between species

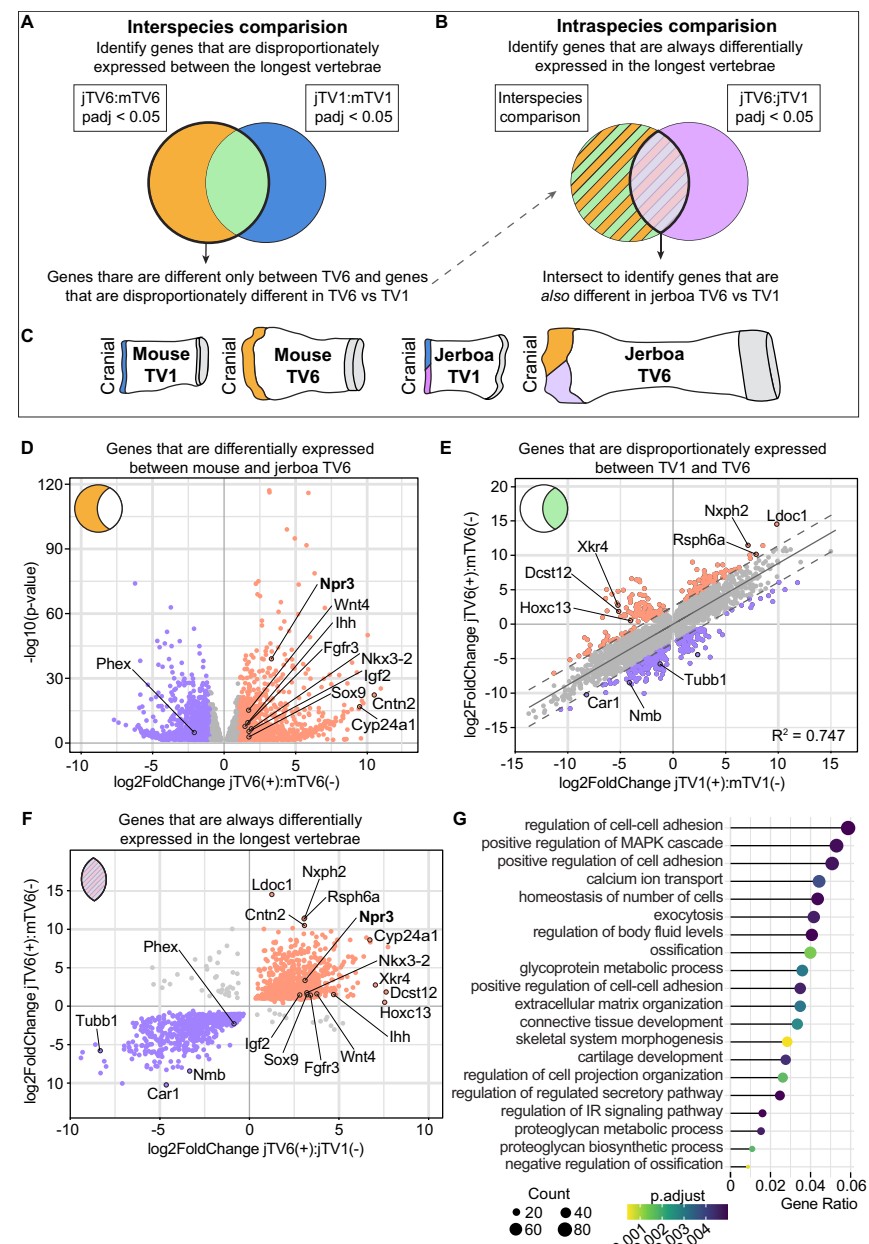

**Fig. 3 | Design and analysis of intersectional interspecies transcriptomics.**
**A**–**C** Schematics of the interspecies and intraspecies comparison approaches of differential expression analyses. **D** 1864 genes are significantly differentially expressed (adjusted *p*-value using Benjamini–Hochberg (BH) method; padj < 0.05) between mouse and jerboa TV6 but not between species in TV1. Genes that are expressed higher in jerboa TV6 are orange; lower are purple. **E** 6786 genes are differentially expressed (padj < 0.05) between jerboa and mouse both in TV6 (*y*-axis) and in TV1 (*x*-axis); most are equivalently differentially expressed (gray points, slope = 0.883), but 421 genes are outside of the 95% prediction interval and

designated "disproportionately differentially expressed". **F** Of all 2285 genes highlighted in (**D** and **E**), 1454 are also differentially expressed in jerboa TV6 versus TV1, consistently in the same direction in the longest versus shortest element. Genes in orange are expressed higher in jerboa TV6 compared to mouse TV6 and in jerboa TV6 compared to jerboa TV1, while those lower in both comparisons are in purple. **G** Selection of GO terms relevant to cartilage that are enriched among the 1454 candidate genes. clusterProfiler uses hypergeometric distribution to calculate gene enrichment, adjusted *p*-values are calculated using the BH method. Source data are provided in Supplementary Data file.

with an additional normalization step to account for gene length differences between species[30,45]. Principal components analysis (PCA) revealed that the variation between samples is primarily due to species differences (PC1) and next by growth cartilage origin (PC2) (Fig. S5).

To narrow our data set to candidate genetic drivers of the evolution of tail proportion, we performed a series of comparisons and intersections of differential expression datasets (Fig. 3A–C). Jerboa TV6 elongates disproportionately faster than mouse TV6, and TV1 is more similar in both species. We therefore directly compared expression between homologous vertebrae (jerboa TV6 to mouse TV6;

jerboa TV1 to mouse TV1) (Fig. 3A). We reasoned that genes that drive the disproportionate elongation of jerboa TV6 should be disproportionately differentially expressed in jerboa TV6. These fall within two categories: 1864 genes are differentially expressed between jerboa and mouse TV6 (padj < 0.05) but not between species in TV1 (Fig. 3E and Supplementary Data 1). We then intersected the log₂-fold change values of all genes differentially expressed (padj < 0.05) between species in both vertebrae (Supplementary Data 2). Most of these are near-equivalently differentially expressed (R² = 0.747; slope = 0.883) and are therefore unlikely to contribute to the

differential elongation of jerboa TV6. Exclusion of genes within the 95% prediction interval of the correlation leaves 421 disproportionately differentially expressed genes, which we added to the genes that significantly differ between species only in TV6 (Fig. 3D and Supplementary Data 3).

We then reasoned that genes that control the disproportionate rate of jerboa TV6 elongation should also differ in their expression between jerboa TV6 and jerboa TV1. Intra-species comparison within jerboa revealed 7911 genes are differentially expressed between vertebrae despite sharing embryonic origins, tail regional identity in the axial skeleton, and developmental age (Fig. 3B and Supplementary Data 4). We intersected these with the genes that are disproportionately differentially expressed between species and found 1454 genes that are differentially expressed in both datasets with the sign value consistently correlating with the most rapid rate of elongation in jerboa TV6 (Fig. 3F and Supplementary Data 5).

GO term enrichment analyses provide valuable insight into possible biological processes regulating disproportionate vertebral elongation, even though knowledge bases are restricted to the fraction of genes that have been studied in limited biological contexts[46]. To identify biological processes enriched in this set of 1454 genes that are differentially expressed within the jerboa and also disproportionately between species, we implemented the clusterProfiler package[47,48]. Among the top significantly enriched terms, many biological processes are critical to cartilage development and ossification (Fig. 3G).

A major goal of this study was to determine if genes that control vertebral proportion overlap significantly with genes associated with limb bone proportion. In addition to our previous work that identified candidate mechanisms of disproportionate jerboa hindlimb elongation[30], an expression study in mice and rats identified genes that are differentially expressed in growth cartilages that elongate at different rates in the same young animal (fast proximal tibia versus slower distal phalanx)[29]. The same study also identified genes differentially expressed in the rapidly elongating young tibia versus the older tibia that slows its growth rate. Although not an equivalent differential expression analysis, a genome-wide association study identified genes near single-nucleotide polymorphisms in a human population that are associated with variance in the proportion of leg length with respect to crown-rump length[28].

We intersected these four datasets with our set of genes associated with the evolution of tail vertebral proportion. A Fisher's exact test with Benjamini–Hochberg multiple hypothesis correction found significant overlap between our gene set and all other datasets (p-values listed in Supplementary Data 6). Therefore, the candidate genes for vertebral proportion are significantly similar to candidate mechanisms of limb bone proportion, even though 80.3% of genes associated with vertebral proportion were not found in any limb proportion study.

To identify genes with the strongest indication of a mechanistic role in the broad regulation of skeletal proportion, we collated genes that also have reported "short tail" or "long tail" mutant phenotypes in the Mouse Genome Informatics (MGI) database (Supplementary Data 6)[49,50]. Of the 1454 genes that correlate with the rapid elongation of jerboa TV6, (Supplementary Data 7), 20 have a reported tail mutant phenotype and/or also appear in three or more limb proportion datasets. Many of these are known regulators of chondrocyte proliferation and/or maturation into hypertrophy. We cannot say how many sequence changes caused skeletal proportion to evolve, though it is likely highly genetically complex, but all of these genes are strong candidates either by their modular cis-regulatory control or by modular expression of upstream transcription factors.

A single gene, Npr3, is common to all five analyses (Table 1 and Supplementary Data 7). Npr3 is more highly expressed in the rapidly elongating jerboa TV6 compared to mouse TV6 and jerboa TV1, higher

in jerboa metatarsals than in mouse[30], and higher in the most rapidly elongating mouse and rat limb bones by age or by location[29]. These data suggest that Npr3 may be a major contributor to the development of proportion broadly in the skeleton and may contribute to shaping its evolution.

## Regulated natriuretic peptide signaling contributes to tail vertebral proportion

NPR3 belongs to a family of receptors that bind C-type natriuretic peptide (CNP) and regulate diverse cellular processes from cardiac function and blood pressure to endochondral ossification[51–53]. NPR2, the primary signaling receptor of this family, is expressed in the growth cartilage and is crucial for normal bone elongation by regulating MAPK signaling through cGMP-mediated activation of PKG[54–56]. Natriuretic peptide receptor C (NPR3) is an inhibitory regulator of natriuretic peptide signaling through clearance-mediated degradation of the natriuretic peptide ligand[57]. Mutations in the NPR3 receptor typically cause skeletal overgrowth, presumably through increased activation of the NPR2 receptor due to excess available natriuretic peptide[55,57–59].

Loss of NPR2 and CNP (Nppc) expression in humans causes Acromesomelic Dysplasia Maroteaux Type, distinguished by disproportionate shortening of the most distal limb elements, implicating NPR signaling in establishing proportion in the limb skeleton[60–67]. However, while inactivating mutations in Npr2 and Npr3 shorten or lengthen the mouse tail, respectively, consistent with overall effects on body size, alterations to vertebral proportion were not documented[52,57,58,60,68,69]. We therefore obtained Npr3 knock-out mice to investigate the impact of loss of NPR3 on overall tail-to-body and individual vertebral proportions.

We collected Npr3[−/−] mice and wild-type siblings at P7 and once they reached adult proportions at P42 and prepared the sacral and caudal tail skeleton for μCT analysis. Compared to mice from our temporal analysis of typical tail growth (CD1 genetic background), total tail length to naso-anal length is not significantly different in mutant or wild-type siblings (C57/Bl6 genetic background) at P7 (Fig. 4A). However, Npr3[−/−] mice have a significantly greater tail to body length ratio at P42, though they also have characteristic but variable spinal kyphosis by this age (Fig. 4A). We then measured lengths of individual tail vertebral centra in mutants and wild-type siblings at P7 and P42 (Fig. 4B).

As in CD1 mice, the "crescendo-decrescendo" pattern is initiated in wild-type siblings at P7. In contrast, vertebral lengths descend from longest to shortest in the Npr3[−/−] mice at P7. Ultimately, both mutants and wild-type siblings have a "crescendo-decrescendo" by P42, but the proximal and mid-tail vertebrae are all disproportionately longer in Npr3[−/−] mice. We speculate that the longer proximal vertebral elements, which otherwise develop normally, may also explain the "typical cone-shaped implantation of the tail" previously documented in Npr3[−/−] mice[58] (see Figs. 4C and S6).

To detect differences in NPR3 expression and localization in mouse and jerboa vertebrae, including Npr3[−/−] and wild-type siblings, we collected TV1 and TV6 cranial growth cartilages during the rapid growth window in each species. NPR3 has previously been reported in the hypertrophic zone of fetal limb growth cartilages[70]. As expected for a null allele, we detected no NPR3 in Npr3[−/−] growth cartilages (Fig. 4D, E). We find NPR3 expression throughout postnatal TV1 and TV6 growth cartilages of wild-type mice (Fig. 4D, E) and in the respective jerboa vertebral cartilages (Fig. 4F, G). Relative to other expression, NPR3 appears more highly expressed in pre-hypertrophic chondrocytes of jerboa TV6 (Fig.4G). We hesitate to compare absolute NPR3 protein expression levels due to potential for detection differences between species.

We then quantified TV1 and TV6 growth cartilage histology in Npr3[−/−] mice and wild-type siblings to determine how loss of NPR3 may

**Table 1 | Twenty candidate genes associated with vertebral and limb proportion**

| Symbol | Name | Dataset | Skeletal phenotype |
|---|---|---|---|
| Crispld1 | Cysteine-rich secretory protein LCCL domain containing 1 | SHR, MY, TP | None noted |
| Cyp26a1 | Cytochrome P450, family 26, subfamily a, polypeptide 1 | TM | Abnormal vertebrae morphology, short tail, kinked tail, vertebral transformation, absent tail[102,103] |
| Fgfr3 | Fibroblast growth factor receptor 3 | TM, TP | Abnormal skeleton development, abnormal vertebrae morphology, short tail, kinked tail, abnormal proliferative and hypertrophic zones[104–106] |
| Hoxc13 | Homeobox C13 | TM | Short tail, caudal vertebral transformation[76,77] |
| Ihh | Indian hedgehog | JH, MY, TP | Abnormal vertebrae morphology, short tail, abnormal endochondral ossification, decreased width of hypertrophic zone, decreased length of long bones[107–109] |
| Lmcd1 | LIM and cysteine-rich domains 1 | JH, MY, TP | Decreased bone mineral content[92,110] |
| Nkx3-2 | NK3 homeobox 2 | TM | Abnormal vertebral morphology, short tail, kinked tail, abnormal axial skeleton morphology[111,112] |
| Nlgn3 | Neuroligin 3 | JH, MY, TP | Increased bone mineral density, abnormal postnatal growth[92,110] |
| Npr3 | Natriuretic peptide receptor 3 | SHR, JH, MY, TP | Long tail, kinked tail, abnormal skeleton development, delayed endochondral ossification, increased width of hypertrophic chondrocyte zone, abnormal vertebrae development[57,58] |
| Pappa2 | Pappalysin 2 | JH, TM | Short tail, decreased body length, decreased length of skeletal elements[113] |
| Phex | Phosphate regulating endopeptidase homolog, X-linked | TM, MY | Short tail, decreased bone mineral density, abnormal hypertrophic chondrocyte zone, small caudal vertebrae, abnormal skeleton morphology[114] |
| Prickle1 | Prickle planar cell polarity protein 1 | JH, TM | Short tail, kinked tail, delayed endochondral ossification, decreased length of skeletal elements[115] |
| Ror2 | Receptor tyrosine kinase-like orphan receptor 2 | TM | Short tail, kinked tail, abnormal cartilage morphology, abnormal vertebra morphology, abnormal skeleton development[116] |
| Sox5 | SRY (sex determining region Y)-box 5 | SHR, TM | Short tail, abnormal bone mineralization, abnormal cartilage, short limbs[117] |
| Sox9 | SRY (sex determining region Y)-box 9 | TM | Short tail, kinked tail, abnormal vertebral morphology, abnormal cartilage development, abnormal skeleton development[118] |
| Thra | Thyroid hormone receptor alpha | JH, TM | Short tail, delayed endochondral ossification, decreased length of skeletal elements[119,120] |
| Ttll9 | Tubulin tyrosine ligase-like family, member 9 | JH, MY, TP | None noted |
| Twsg1 | Twisted gastrulation BMP signaling modulator 1 | TM | Short tail, kinked tail, decreased caudal vertebrae number, abnormal skeleton morphology, abnormal cartilage, decreased length of skeletal elements[121,122] |
| Wnt4 | Wingless-type MMTV integration site family, member 4 | SHR, MY, TP | Abnormal cartilage morphology, decreased chondrocyte proliferation[83] |
| Zim1 | Zinc finger, imprinted 1 | JH, MY, TP | None noted |

Mutant phenotypes identified in MGI were confirmed in primary literature. JH—candidate genes driving jerboa hindlimb elongation[26]. MY—genes that are differentially expressed in the rapidly elongating young tibia versus the mature tibia[25]. TP—genes that are differentially expressed in the fast-growing proximal tibia versus slow distal phalanx[25]. SHR—SNPs associated with variance in the proportion of leg length with respect to total body length[30]. TM—genes with reported "short tail" or "long tail" mutant phenotypes in the MGI database[31,32].

drive the disproportionate elongation of proximal tail vertebrae at the cellular level (Fig. 5A–F). We found that growth cartilage heights are significantly taller in both TV1 and TV6 of mutant compared to wild-type siblings (Fig. 5C). However, proliferative zone height is only significantly different between mutant and wild-type in the caudal TV1 growth cartilages (Fig. 5D). In contrast, the hypertrophic zone is significantly taller at both cranial and caudal ends of TV1 and TV6 in Npr3$^{-/-}$ compared to wild-type, particularly at TV6 where the mutant hypertrophic zone is nearly twice the height of the wild-type hypertrophic zone (Fig. 5E). To determine if this is due to an increase in hypertrophic chondrocyte size, we measured the average maximum diameter of hypertrophic chondrocytes in the axis of elongation in each growth cartilage. We found that hypertrophic cell size is approximately 20% greater in the Npr3$^{-/-}$ growth cartilages, and that hypertrophic cells are similarly sized between mutant TV1 and TV6 (Fig. 5F). Collectively, these data suggest that disproportionate elongation of the proximal tail vertebrae in Npr3$^{-/-}$ mice is at least in part due to expansion of the hypertrophic zone through an increase in cell size.

These data demonstrate that natriuretic peptide signaling regulates chondrocyte differentiation and hypertrophy to control vertebral proportion in addition to the role it plays in limbs. Differential expression of Npr3 in our RNA-seq data suggests that this pathway may also be one of many genetic mechanisms that drove the evolution of the jerboa mid-tail region and total tail length.

## Discussion

Here, we used the remarkable diversity of the axial skeleton, both within an individual and between species, to study how serially repeating elements reach distinctly different sizes. We provide valuable insights into how vertebrae that are considered to have the same regional identity elongate at different rates. Our intersectional approach, comparing vertebrae with disproportionate (TV6) to those with similar growth rates (TV1), allowed us to identify cellular mechanisms and candidate genes associated with disproportionate vertebral elongation.

Mammal tails differ in both the number and lengths of individual vertebrae[3,26,27,71]. Consistent with the different developmental mechanisms determining vertebral number (somitogenesis) and relative lengths (endochondral ossification), these characters are also genetically separable. In deer mice (Peromyscus maniculatus), six genomic regions (loci) contribute to total tail length differences between forest and prairie ecotypes. Three contribute to differences in vertebral number, but not length, and three to length but not number[27].

Research in laboratory models (mice and chickens) has shown that somitogenesis, which determines vertebral number, is partly controlled by genes in an interaction network[72,73]. Out of 168 genes in the deer mouse "vertebral number" loci, prior knowledge of this network focused attention on differences in the non-coding control of

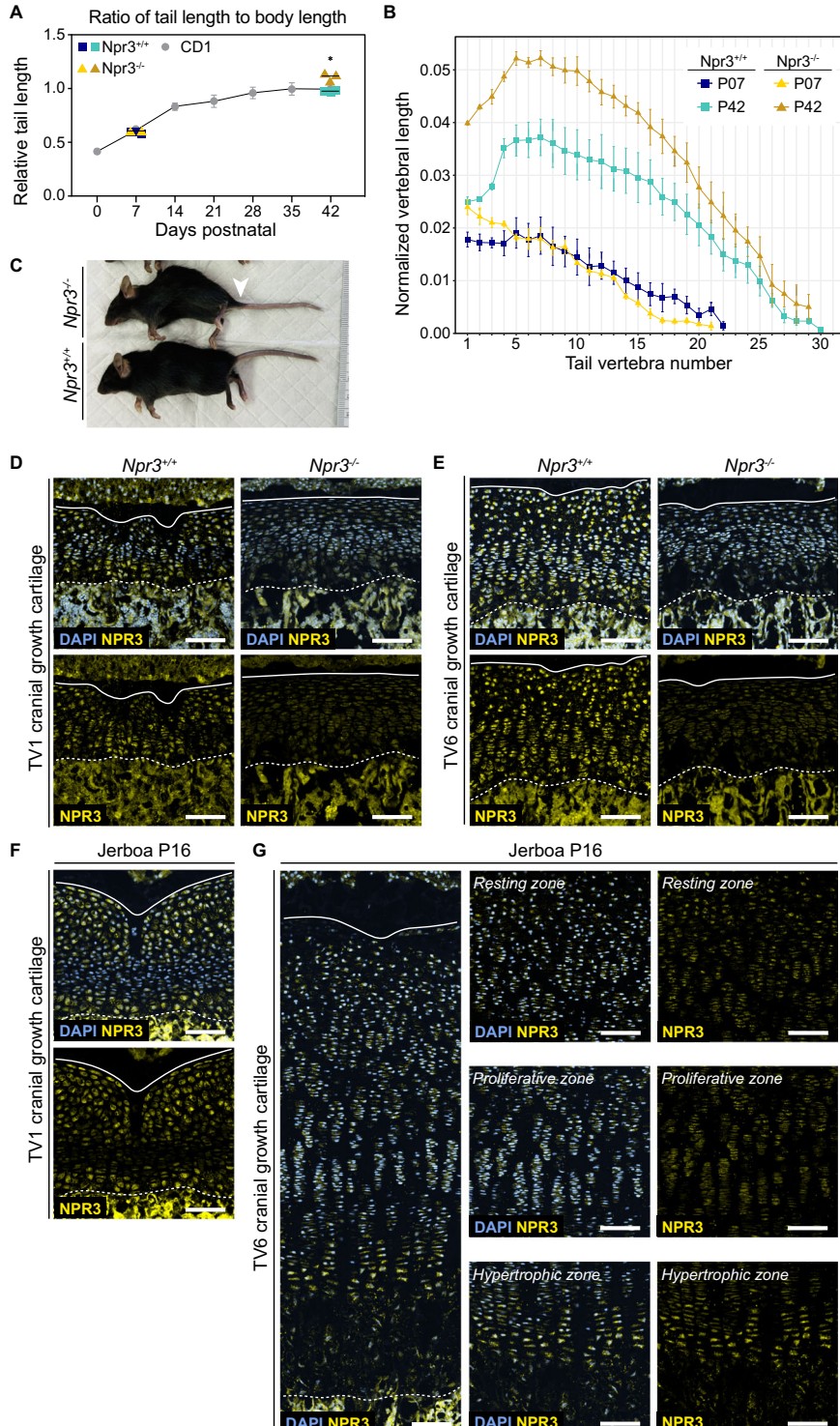

**Fig. 4 | Natriuretic peptide signaling affects mouse tail vertebrae proportion.**
**A** Ratio of tail length to naso-anal distance in postnatal (P7) and adult (P42) Npr3⁻/⁻ mice (triangles, yellow/gold) and wild-type Npr3⁺/⁺ siblings (squares, blue/turquoise), where solid lines indicate means, compared to average relative tail lengths in CD1 mice (gray circles, error bars show the standard deviation). Two-way ANOVA performed for P42, $p = 0.0491$. **B** μCT scans were used to measure the lengths of vertebral centra in the same animals at P7 (yellow; blue) and P42 (gold; turquoise), normalized to the naso-anal length of each Npr3⁻/⁻ (triangles) and wild-type Npr3⁺/⁺ siblings (squares). Points represent the mean vertebral lengths at each position and timepoint with error bars showing the standard deviation of these measurements. For Npr3⁺/⁺ and Npr3⁻/⁻ P42 $n = 3$ of mixed male and female animals, for CD1 P42 $n = 3$ males and 3 females. Due to one mutant having a tail

kink, $n = 2$ for Npr3⁻/⁻ at P7 for total tail proportion. **C** Npr3⁻/⁻ mouse with wild-type sibling aged P42. White arrowhead points to "cone-shaped implantation of the tail". **D** Immunofluorescent expression of NPR3 in Npr3⁻/⁻ and Npr3⁺/⁺ TV1 and (**E**) TV6 cranial growth cartilages at P7. **F** Immunofluorescent staining of NPR3 in jerboa TV1 and **G** TV6 cranial growth cartilages at P16. For each of Npr3⁺/⁺ and Npr3⁻/⁻, $n = 3$ of mixed male and female animals. The panel in (**G**) has been subdivided into growth zone insets to show expression in resting, proliferative, and hypertrophic zones. Solid white lines show the edge of the intervertebral disc and dotted white lines demark the chondro-osseous junction. DAPI stain is shown in cornflower blue and NPR3 in yellow. Scale bar is 100 μm. Source data are provided as a Source Data file.

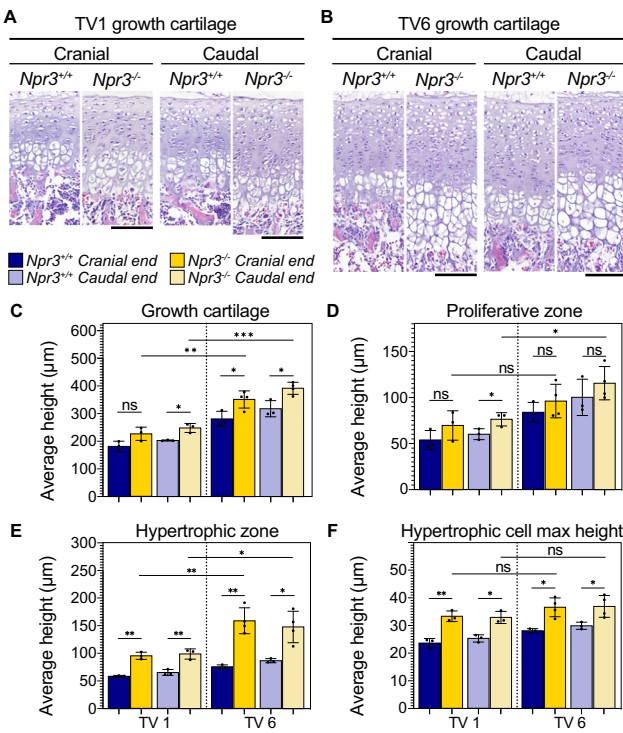

**Fig. 5 | Loss of NPR3 drives rapid cartilage elongation in part by increasing chondrocyte hypertrophy. A**, **B** Histology of mouse Npr3+/+ and Npr3−/− cranial and caudal growth cartilages of TV1 (**A**) and TV6 (**B**) during rapid elongation at P7. Scale bar is 100 μm. **C–F** Growth cartilage parameters in each cranial and caudal cartilage of Npr3+/+ and Npr3−/− TV1 and TV6. **C** Height of the growth cartilage. **D** Height of the proliferative zone. **E** Height of the hypertrophic zone. **F** Maximum height of hypertrophic chondrocytes in the direction of bone elongation. Welch's *t*-test, two-tailed, *n* = 3 for all Npr3+/+ cartilages and Npr3−/− TV1, *n* = 4 for Npr3−/− TV6 cartilages, mixed male and female animals, * ≤ 0.05, ** ≤ 0.01, *** ≤ 0.001. Exact *p*-values are available in Source Data file. **C–F** Error bar centers are the data means and error bands show standard deviation. Source data are provided as a Source Data file.

*HoxD13* as potentially causative[26,27]. However, details of the hundreds of genes in the "vertebral length" loci remain unreported, mainly because less was known to winnow the list down to likely functional candidates. Therefore, our analyses of the cellular and genetic differences driving differential growth within and between jerboas and mice are a crucial first step to identify causative mechanisms of vertebral length diversification in other mammals.

Despite the sizable gap in understanding vertebral elongation, we have a rich knowledge of limb bone growth from decades of studies. However, even though both limb and vertebrae elongate by endochondral ossification of a growth cartilage, the limb and vertebral skeletons have distinct developmental origins. Furthermore, the vertebral endochondral skeleton likely pre-dates tetrapod limbs by at least 60 million years[19–21]. It was therefore entirely unclear whether the cellular and molecular mechanisms that fine-tune local growth to establish and diversify proportion should be similar or distinct.

Total growth cartilage height corresponds strongly with growth rate in bird, reptile, and mammal limb bones; we show here that growth cartilage height also associates with tail vertebral growth rate in mice and jerboas. This suggests that growth cartilage height and cell number are readily evolvable in the limb and vertebral skeleton, possibly traceable to the origin of tetrapods. In mammal, but not in bird or reptile limb bones, hypertrophic chondrocyte size differs significantly and contributes the most to differences in the daily rate of limb bone

elongation within and between many mammal species[33,34,37–41]. By contrast, we show that hypertrophic chondrocytes are the same size in mouse vertebrae elongating at different rates and that these are smaller than limb bone hypertrophic chondrocytes. Hypertrophic chondrocytes are only larger in the most rapidly elongating jerboa tail vertebrae, suggesting that cell size differs widely in mammal limbs, but perhaps only in vertebrae when a greater difference may contribute to extreme vertebral proportion.

Identifying the genetic mechanisms of macroevolution over many millions of years is undoubtedly a quantitative genetics (polygenic) problem and is thus unamenable to classical genotype-phenotype approaches (see Cooper 2024[74]). For example, the population diversity of human height, a continuous trait, is associated with thousands of genetic variants that likely each have a very small effect (recently reviewed in Bicknell 2025[75]); the macroevolution of skeletal proportion is likely similarly complex. However, even if full understanding of the complex regulatory mechanisms of evolution is further from reach, we can identify candidate genes and molecular networks driving relevant biological processes and use these data to understand fundamental mechanisms necessary to establish proportion.

Here, we show significant overlap of differentially expressed genes in our vertebral comparisons, three limb differential expression datasets, and a human GWAS analysis of limb to trunk proportion, suggesting some of the same mechanisms control proportion in the limbs and vertebrae. However, despite the significant overlap, a majority (80.3%) of the differentially expressed genes in tail vertebrae are not among limb proportion genes. Genes among these might therefore control vertebral proportion independent of the limb skeleton. For example, *HoxC13* regulates posterior embryonic development and segment identity[76]. Intriguingly, loss of *HoxC13* causes distal TV9–11 to acquire shapes that are typical of more proximal vertebrae, TV4-6 (transverse processes, see Fig. 1C), and the longest vertebra shifts distally; conversely, *HoxC13* overexpression reduces total tail length[76,77]. These may be important observations to answer the larger question of how neighboring vertebrae differentiate in shape and size from one another within a shared regional identity. Because *HoxC13* seems to disproportionately affect the proximal tail, it may play a role in integrating signaling molecules (e.g., NPR peptides) and cellular contexts differently in neighboring tail vertebrae.

Although most differentially expressed genes are unique to the vertebrae, those shared with limbs suggest that central components of the chondrocyte maturation process are locally tuned to regulate development and evolution of proportion throughout the skeleton. Among these shared genes, we find *Ihh*, *Wnt4*, and *Fgfr3*. *Ihh* regulates chondrocyte proliferation and hypertrophy in a negative feedback loop with *PTHrP*, promoting proliferation and maintaining bone growth while indirectly delaying hypertrophy[78–82]. In contrast, *Wnt4* stimulates chondrocyte hypertrophy, and overexpression of this gene reduces chondrocyte proliferation, because maturation is accelerated[83,84]. During postnatal growth, *Fgfr3* is expressed in the proliferative zone and suppresses chondrocyte proliferation and hypertrophy via activation of STAT and MAPK/ERK signaling, respectively[85–88].

Activating mutations in the *Fgfr3* receptor are the most common cause of human achondroplasia[89]. Because of pathway interactions between FGF and natriuretic peptide receptor (NPR) signaling[53,55,56], Vosoritide, a CNP analog, is a relatively new intervention to treat complex aspects of the syndrome[56,90,91]. Interestingly, natriuretic peptide receptor C (*Npr3*) is the only gene we found in all limb and vertebral differential expression datasets, human body proportion GWAS, and mouse tail mutant phenotypes, suggesting it plays a central role in establishing and perhaps also diversifying skeletal proportion.

Paradoxically, although NPR3 loss-of-function mice have disproportionately longer mid-tail vertebrae, NPR3 expression is consistently higher in the fastest-growing limb bones of mice, rats, and

jerboas, as well as jerboa TV6, compared to bones that elongate at a slower rate. NPR3 is thought to suppress NPR2-mediated promotion of bone elongation. NPR2 and NPR3 are detected throughout the growth cartilage; NPR2 is most expressed in the resting and proliferative zones, while NPR3 expression is most pronounced in the hypertrophic zone[70]. Altered NPR3 expression may therefore have zone-specific functions not evident in the constitutive loss-of-function phenotype. Further, in addition to its role as a ligand clearance receptor, NPR3 has been reported to function as a signaling receptor by modulating levels of cAMP and downstream PKA activity in the cranial placode[59]. Therefore, NPR3 may have additional roles in regulating long bone elongation besides CNP clearance. Alternatively, in a polygenic system that evolved over tens of millions of years, higher expression of NPR3 might constrain even greater elongation promoted by other genetic differences to reach an adaptive balance. Regardless of this remaining uncertainty, our data support the hypothesis that natriuretic peptide signaling is an important regulator of vertebral skeletal proportion by modulating growth plate height, particularly in the hypertrophic zone, and hypertrophic chondrocyte size[57].

Several genes appearing in multiple skeletal proportion datasets do not have a known or clear role in cartilage development. Four of these genes (*Lmcd1, Nlgn3, Ttll9,* and *Zim1*) that we identified in the disproportionately elongated jerboa TV6 and jerboa metatarsals are also among differentially expressed genes in the mouse and rat limb proportion study[29]. While mutations in *Lmcd1* and *Nlgn3* are reported to cause abnormal bone mineral density and postnatal growth defects[92,93], *Ttll9* and *Zim1* have no reported bone defects and have not been investigated in the growth cartilage.

Here, we have identified cellular mechanisms of differential growth in the vertebral skeleton and candidate genetic regulators of proportion. Some of these may specifically control vertebral proportion, while others suggest common mechanisms locally regulate growth rates throughout the skeleton. This opens avenues to focus attention on mutations that diversify proportion in natural populations, to identify *cis*-regulatory elements that modularize the timing and level of expression in individual growth cartilages to achieve different growth rates, and to test the functions of genes not previously assigned a role in skeletal elongation.

## Methods

All animal care and use protocols were approved by the Institutional Animal Care and Use Committee of the University of California, San Diego. All key reagents and resources referenced in the following methods are also listed in Table 2.

### Experimental model and subject details

Both mouse and jerboa vivarium adhere to a 10-h light/14-h dark cycle with room temperature set to 70 °F ± 4 and humidity set to ~55%. Jerboa-specific housing has been described previously described[94]. CD-1 and C57BL/6 mice were obtained from Charles River Laboratories (MA, USA). We tested the effect of sex on relative tail length in CD-1 mice (Fig. 1D) and found no significant difference at any time point from P0 to P42. Male and female animals were therefore randomly assigned for subsequent histological analyses. NPR3 mutant mice (MGI:2158355) were obtained from Dr. Jeffery Olgin's laboratory at University of California, San Francisco. Male and female *Npr3*$^{-/-}$ mice were initially crossed to C57BL/6 mice to expand the line. Due to the effect of NPR signaling on female ovarian biology and male penile function, both mutant and heterozygous animals up to 6 months of age were used for breeding to obtain mutant animals for study.

### Skeletal preparations and image capture

Adult mice and jerboas were humanely euthanized, then carcasses were skinned and placed in an enclosed colony of dermestid beetles. Once the skeleton was cleared of tissue, but joint articulations were still intact, skeletons were brushed off then frozen to kill surviving beetles/larvae. Skeletons were further cleaned by hand and brightened in a gentle 5% hydrogen peroxide solution.

Neonates were humanely euthanized, then carcasses were skinned and eviscerated and fixed in 95% ethanol overnight. Samples were stained overnight in cartilage staining solution (75% ethanol, 20% acetic acid, and 0.05% Alcian blue 8GX), rinsed in 95% ethanol, cleared overnight in 0.8% KOH, then stained again overnight in bone staining solution (0.005% Alizarin red S (Sigma-Aldrich, A5533) in 1% KOH). Specimens were cleared in 20% glycerol in 1% KOH until the entire axial skeletal morphology was visual to the eye. For imaging and storage, samples were processed through a series of glycerol in 1% KOH up to 100% glycerol. Protocol modified from Lim et al. 2024[95].

### Carcass preparation and uCT analysis
**Carcass preparation.** We performed transcardiac perfusion fixations to prepare entire carcasses for micro-computed tomography imaging. Animals were terminally anesthetized with a ketamine/xyaline solution. Once sufficient anesthesia was reached, we opened the rib cage to expose the heart and inserted a 21-gauge butterfly needle attached to a 50 cc syringe with warmed sterile saline into the left ventricle and cut a small incision into the right atrium to allow flow through. Saline was slowly flushed until the liver lost its red color and the solution flowed clear. The syringe was then exchanged for a 50 cc syringe containing freshly prepared 4% PFA, which was slowly injected until muscle twitching followed by limb stiffness was observed. Animals were then transitioned into 70% EtOH for storage. We collected ≥3 animals for each time point and experiment (see Table 3). Both males and females were collected at each time point.

**µCT image collection and analysis.** CD1 mice and jerboas were scanned at 9 µm³ voxel resolution on a Skyscan 1076 MicroCT machine (50 kVp, 200 µA, 0.5 mm aluminum filter, 180° scan, Δ = 0.8°). Specimens were supported during scanning using plastic pellets. Images were reconstructed using NRecon (Bruker, Belgium) with a smoothing factor of 1, ring artifact reduction factor of 6, beam hardening correction factor of 40%, and with a dynamic range from 0 to 0.11 attenuation units. Each reconstructed dataset was immediately viewed in Dataviewer to assess the quality of each scan. Due to a defect in the scanner that arose late in the experimental progress of this manuscript, some individuals were scanned twice (A/B numbered scans) so that all vertebrae could be measured without interference. NPR3 mice were scanned on a Revvity Quantum GX2 microCT imaging system (Clemson, SC, USA) at 20 µm voxel resolution, 90 kV voltage, 88 mA current., Cu 0.06 + Al 0.5 filter, 360° scan).

Whole µCT files were opened for analysis in Bruker DataViewer software. Due to the size of the files, CD1 and jerboa scans were always resized by a factor of 3. NPR3 animals were not resized. Skeletons were first oriented to the first sacral vertebra, which is easily identifiable due to its unique morphology. The vertebral centrum was aligned in all axes before measuring the diaphyseal bone length in each element in the sagittal plane in sequence to the tail tip. The growth plates, including the epiphyses, were not included in measurements, because epiphyses have not formed by birth, and joint interzones cannot be detected by µCT.

To make a measurement, the cursor was drawn from the outside of the cranial end of the diaphysis in a straight line to the caudal end. Pixel intensities were measured over this distance and saved in an individual.csv file for each vertebra. The length of the vertebral diaphysis was determined to begin at the midpoint between the first pixel intensity minimum/maximum and to end at the midpoint of the last maximum/minimum. Pixels were converted to microns based on the scan resolution and resizing factor to determine the length of each element. Vertebral lengths were normalized to the naso-anal distance

## Table 2 | Key resources table

| Reagent/Resource | Source | Identifier |
|---|---|---|
| **Chemicals, peptides, etc** | | |
| Alcian 8GX *certified by Biological Stain Commission* | Sigma-Aldrich | Cat# A3157 |
| Alizarin red S | Sigma-Aldrich | Cat# A5533 |
| OCT compound | Tissue-Tek | Ref# 4583 |
| Proteinase K | Qiagen | Cat# 19131 |
| MyTaq Red PCR mix | Bioline | Cat# BIO-25047 |
| Calcein | Sigma-Aldrich | Cat# C0875 |
| Eosin-Y | Thermo scientific | Cat# 201931000 |
| Hematoxylin | Sigma-Aldrich | Cat# GHS132 |
| CryoJane tape-transfer system | Leica | |
| Tape windows | Leica | Cat# 39475214 |
| Solution A | Leica | Cat# 39475270 |
| Solution B | Leica | Cat# 39475272 |
| Fluoromount-G mounting medium | Invitrogen | Cat# 00-4958-02 |
| Bioworld epitope unmasking buffer | Fisher scientific | Cat# 50-199-077 |
| NPR3 anti-Rabbit polyclonal antibody | Invitrogen | Cat# PA5-96947 |
| Alexa Fluor 594 Goat Anti-Rabbit IgG (H&L) | Invitrogen | Cat# ab150080 |
| DAPI | Invitrogen | Cat# D1306 |
| **Critical commercial assays** | | |
| Click-iT™ EdU Kit, Alexa Fluor 647 | Invitrogen | Cat# C10340 |
| RNeasy Micro Kit | Qiagen | Cat# 74004 |
| QIAshredder | Qiagen | Cat# 79656 |
| **Experimental models: Organisms/strains** | | |
| *Mus musculus* (CD-1 Strain) | Charles River Labs | Strain Code: 022 |
| *Mus musculus* (C57BL/6 Strain) | Charles River Labs | Strain Code: 027 |
| *Mus musculus* Npr3$^{-/-}$ | Olgin Lab, UCSF | MGI:2158355 |
| *Jaculus jaculus* | UCSD Cooper Lab | |
| **Oligonucleotides** | | |
| NPR3 primer Forward | | 3' CACAAGGACACGGAATACTC 53' |
| NPR3 primer Reverse | | 3' CTTGGATGTAGCGCACTATGTC 53' |
| NPR3 primer Neo Forward | | 3' ACGCGTCACCTTAATATGCG 53' |
| **Software and algorithms** | | |
| FIJI[97] | | https://imagej.net/software/fiji/ |
| InteredgeDistance Macro for FIJI | Santosh Patnaik | |
| DataViewer | Bruker | |
| DESeq2[45] | | https://github.com/thelovelab/DESeq2 |
| STAR[100] | | https://github.com/alexdobin/STAR |
| TOGA[44] | | https://github.com/hillerlab/TOGA |
| CutAdapt[98] | | https://github.com/marcelm/cutadapt |
| FastQC[99] | | https://github.com/s-andrews/FastQC |
| Trim Galore | | https://github.com/FelixKrueger/TrimGalore |
| R packages used in analysis | | |
| ggplot2 | | https://github.com/tidyverse/ggplot2 |
| dplyr | | https://github.com/tidyverse/dplyr |
| Plotly | | https://github.com/plotly |
| clusterProfiler[47] | | https://github.com/YuLab-SMU/clusterProfiler |
| Hrbrthemes *version 0.8.7* | | https://github.com/hrbrmstr/hrbrthemes |
| Paletteer *version 1.3.0* | | https://github.com/EmilHvitfeldt/paletteer |
| BiocManager *version 1.30.25* | | https://github.com/Bioconductor/BiocManager |

## Table 2 (continued) | Key resources table

| Reagent/Resource | Source | Identifier |
|---|---|---|
| GeneOverlap[101] *version 1.40.0* | | https://github.com/shenlab-sinai/GeneOverlap |
| gprofiler2 *version 0.2.3* | | https://github.com/egonw/r-gprofiler2 |
| Tidyr *version 1.3.1* | | https://github.com/tidyverse/tidyr |
| RColorBrewer *version 1.1-3* | | https://github.com/cran/RColorBrewer |
| Viridis *version 0.6.5* | | https://github.com/sjmgarnier/viridis |
| Reshape2 | | https://github.com/hadley/reshape |
| Graphpad Prism software | | |
| Adobe Creative Cloud (Illustrator and Photoshop) | | |

## Table 3 | Number of female (F) and male (M) animals collected for µCT imaging

| | P0 | P7 | P14 | P21 | P28 | P35 | P42 |
|---|---|---|---|---|---|---|---|
| **Mouse F** | 3 | 3 | 4 | 3 | 4 | 3 | 3 |
| **Mouse M** | 3 | 3 | 2 | 3 | 2 | 3 | 3 |
| **Jerboa F** | 2 | 4 | 5 | 1 | 2 | 2 | 3 |
| **Jerboa M** | 3 | 2 | 1 | 5 | 4 | 4 | 2 |
| **Mouse F NPR3$^{+/+}$** | | 2 | | | | | 0 |
| **Mouse M NPR3$^{+/+}$** | | 1 | | | | | 3 |
| **Mouse F NPR3$^{-/-}$** | | 2 | | | | | 2 |
| **Mouse M NPR3$^{-/-}$** | | 1 | | | | | 1 |

per each animal. Normalized vertebral lengths and relative change in vertebral length per week were visualized using ggplot2[96].

### Growth cartilage tissue histology

**Calcein pulse to measure growth rate.** Mice and jerboas received an intraperitoneal injection of 15 mg/kg calcein. Tails were collected exactly 48 h after injection and fixed overnight in cold 4% paraformaldehyde. Tails were then transitioned through a sucrose gradient into Optimal Cutting Temperature (OCT) media, dissected into two segments for sectioning: sacral 4 (S4) to TV2 and TV2 to TV8. Samples were then flash frozen in OCT media in block molds. Blocks were serially tape sectioned using Leica tape sectioning solution and CryoJane materials at 50 micron thickness. For imaging, slides were thawed and gently rinsed in PBS to remove OCT and tape residue. Sections were incubated with DAPI in PBS for 2 min, then mounted in Fluoromount-G Mounting Medium. Sections were imaged on an inverted Olympus Fluoview 3000 laser scanning confocal microscope at 20× magnification. To capture the entire calcein front, and not pieces of the trabeculae, 10 optical sections were captured across 20 microns oriented around the center of the section. Optical sections were visualized in a maximum intensity projection and the distance between the calcein front and chondro-osseous junction, was quantified by spline interpolation using the Inter-edgeDistance Macro (Santosh Patnaik) in FIJI[97] with default settings. Distances were averaged across three sections per individual for every growth plate and visualized in Graphpad Prism.

**EdU Click-iT kit to quantify proliferation index.** Mice and jerboas received an intraperitoneal injection of EdU per the manufacturer's protocol and were collected exactly 2 h after injection. Tails were fixed and prepared for cryosectioning as described above. Blocks were tape sectioned using Leica tape sectioning solution and CryoJane materials at 10 micron thickness. For imaging, slides were thawed and gently rinsed in PBS to remove OCT and tape residue. EdU was then detected using the EdU Click-iT protocol, and slides were mounted in Fluoromount-G Mounting Medium for imaging on an inverted

Olympus Fluoview 3000 laser scanning confocal microscope. Three sections were imaged per individual for each growth plate.

Images were cropped to an ROI that captured the whole proliferative zone as determined by Hoechst co-stain and refined to the top/bottom-most EdU positive cell. Using FIJI, images were thresholded in each channel to create a mask. Watershed function was used to separate adjacent nuclei and "Analyze Particles" function was then used to count each size-restricted shape. To calculate the proliferative index, the ratio of EdU positive cells to total number of cells per ROI was averaged across three sections for each individual. Results were visualized in Graphpad Prism.

**Hematoxylin and Eosin staining for growth plate histology.** Cryo Jane tape sections of 10 micron thickness were used for H&E staining of wild-type mouse and jerboa tails. Paraffin sections of 5 micron thickness were used for H&E staining of NPR3 mouse tails. All sections were stained using the University of Rochester Center for Musculoskeletal Research core H&E protocols for frozen and paraffin sections. Three sections per individual for each growth plate were imaged on an Olympus BX61 upright compound microscope. Spline interpolation was used to measure the height of the entire growth plate, hypertrophic, proliferative zone, and resting zone based on cell morphology. Average maximum cell heights were measured in the axis of elongation through the three largest cells with a clear nuclear profile and averaged across three sections. Results were visualized in Graphpad Prism.

**Immunofluorescence staining.** Raw images were opened in FIJI and prepared for bulk processing in Adobe Photoshop by setting LUTs and separating channels. Channels were opened as layers in Photoshop and grouped by channel (DAPI and NPR3), which were each set as screens against a black background. The DAPI layer group was set to 90% opacity. The Levels tool was used to adjust image brightness, contrast, and tonal range equally for the NPR3 group. Finally, a Brightness layer was imposed across all layers. After batch processing, IF images were moved into Adobe Illustrator for figure construction.

Cryo-tape sections at 10 micron thickness were used for immunofluorescence staining. Slides were rinsed in PBS before antigen retrieval using Bioworld Epitope Unmasking Buffer (citric acid-based) per the manufacturer's protocol. After antigen retrieval, slides were again rinsed in PBS before blocking for 1 h at room temperature (Block: 0.3% BSA, 1% FBS, 0.01% TritonX100). Slides were incubated in NPR3 antibody (1:500) overnight at 4 °C. The next day, slides were rinsed 3 times for 20 min each in PBST and incubated with a secondary antibody (1:1000) overnight at 4 °C. Sections were imaged on an inverted Olympus Fluoview 3000 laser scanning confocal microscope. Laser detection thresholds were set against a no-primary antibody control and the mouse and jerboa TV6 sections and used to image all sections. For each section, 5 z-stacks 2 microns apart were captured.

**RNA sequencing and analysis**
Tails were dissected in ice cold PBS and then incubated in Proteinase K for 10 min as described previously to remove tendons and connective tissue[30]. Tails were then washed in ice cold PBS and then TV1 and TV6 cranial growth plates were microdissected from the tail by cutting at the intervertebral disc and chondro-osseous junction. As much IVD tissue as possible was cut away, but some annulus fibrosus remained in all samples. Three cranial growth plates per vertebral identity per species was pooled into RNAlater (Invitrogen) and equilibrated overnight at 4 °C. Samples were removed from RNAlater and snap frozen in liquid nitrogen the next day and stored at −80 °C or immediately processed for RNA extraction.

For tissue disruption, growth plates were ground into a powder using a mortar and pestle over liquid nitrogen, then homogenized using the Qiagen QIAshredder system per the manufacturer's protocol. RNA was extracted from the tissue homogenate using the Qiagen

RNeasy Micro kit following the manufacturer's protocol, including the on-column DNA digest. Initial RNA concentrations were recorded on a nanodrop before submission to the UCSD IGM core for tape station measurements. Only samples with RIN scores >8 were used for library preparation. UCSD IGM used Illumina mRNA Stranded Library kits to prepare libraries. Samples were run on PE100 lanes to get 25 million reads per sample on an Illumina NovaSeq 6000 platform.

Quality control analysis and trimming of raw fastq files were performed using Trim Galore, a wrapper around Cutadapt[98] and FastQC[99]. Trimmed reads were mapped to TOGA-annotated one-to-one and one-to-zero transcripts using STAR[100], and read counts per gene were acquired by "–quantMode GeneCounts" option. STAR aligned gene counts were used to perform differential expression analysis between mouse and jerboa TV1 and TV6 with DESeq2[45]. To account for variable orthologous/transcript lengths between species we normalized gene lengths in the DESeq2 pipeline as described in Saxena et al., 2022[30]. Default DESeq2 settings were used to perform PCA to identify variance associated with our tissue and species comparisons ($n = 4$ for all species/tissues). DESeq2 differential expression analysis was performed using the Wald test, and the DESeq function performed $log_2$ fold-change shrinkage by default. We considered all differentially expressed genes with an adjusted $p$-value < 0.05 (padj; calculated using the Benjamini−Hochberg method) to be statistically significant in our analyses. R was used to perform all subsequent analyses and all generated code will be available on Dryad (doi:10.5061/dryad.70rxwdc92).

The GO term enrichment analysis was performed using the clusterProfiler package[47,48]. We used the following settings: enrichGO function, org.Mm.eg.db mouse genome, SYMBOL keytype, BP (biological process) subontology (ont), all genes expressed across all growth plates in this analysis constituted the universe, and p-value cutoff was set at 0.05. clusterProfiler uses hypergeometric distribution to calculate gene enrichment.

**Quantification and statistical analysis**
To test the effect of sex on tail elongation, we performed a two-way ANOVA on mouse male and female individual relative tail lengths at each time point and found that there is no significant difference between male and female relative tail length. To test the effect of species on tail elongation, we performed Welch's $t$-tests at each time point between mouse and jerboa and found that relative tail lengths are not significantly different from P0 to P14, and jerboa tails become significantly longer from P21 onwards.

We used the Shapiro−Wilk and Kolmogoroc−Smirnov tests to determine the normality of our histological measurement data and found that not all parameters are normally distributed, thus we performed an unequal variances $t$-test or Welch's $t$-test to compare means between groups. We performed Welch's $t$-test to compare cranial/caudal growth cartilages within species (i.e., mouse cranial TV1 vs. mouse cranial TV6) or growth cartilages of a single vertebra (i.e., mouse cranial TV1 vs. mouse caudal TV1).

We used the GeneOverlap package[101] in R to perform Fisher's exact tests with Benjamini−Hochberg correction to compare the overlap between our 1454 candidate genes and other datasets investigating skeletal proportion.

**Reporting summary**
Further information on research design is available in the Nature Portfolio Reporting Summary linked to this article.

## Data availability
The RNA sequencing data generated in this study have been deposited in the GEO database under accession code GSE299515. The processed RNA-seq data, 1:1 and 1:0 gtf annotation files generated with TOGA as well as gene lengths for these annotations, are available on Dryad

[https://doi.org/10.5061/dryad.70rxwdc92]. Raw images and histological measurements are also available on Dryad [https://doi.org/10.5061/dryad.70rxwdc92]. Normalized measurements and gene subsets used to generate the graphs in Figs. 1–5, S1, and S4 are provided in the Supplementary Data and Source Data files. Due to the size of the raw μCT files, these data are not yet available on a public repository. We have uploaded compressed files to a read only OneDrive and will grant access to data upon request. Source data are provided with this paper.

## Code availability

The R code scripts used to run DESeq2 and analyze RNA-seq differential expression are available on Dryad [https://doi.org/10.5061/dryad.70rxwdc92]. The Python script written by author A.J.W. to automate analysis of μCT raw measurements is also available on Dryad [https://doi.org/10.5061/dryad.70rxwdc92].

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

## Acknowledgements

We are grateful to all members of the Cooper laboratory for assisting in these experiments, especially Daniel Ochoa-Reyes, Sara Kamenski, Sarah Pratt, and Jolie Adair, and for discussing data and interpretations. We thank Jeffrey Olgin at UC San Francisco for providing the *Npr3* knockout mice and Eric Chang, Saeed Jerban, and Qingbo Tang at the VA Medical Center in San Diego for using their microCT scanner. We also thank Talia Moore, Terry Capellini, Matt Hilton, Richard Behringer, and Hopi Hoekstra for their mentorship and support of C.J.W. and Andrew McCulloch and Sam Ward for insightful discussion. This publication includes data generated at the UC San Diego IGM Genomics Center utilizing an Illumina NovaSeq 6000 purchased with funding from a National Institutes of Health SIG grant (#S10 OD026929). We thank Kristen Jepsen at the UCSD IGM for her help and guidance in designing and executing the sequencing experiments. This work was supported by the National Institutes of Health Ruth L. Kirschstein National Research Service Award (NRSA) Individual Postdoctoral Fellowship award number F32AR079923 to C.J.W. and the National Institutes of Arthritis, Skin, and Musculoskeletal Diseases award number R01AR075415 and by the National Science Foundation under grant number IOS-2500299 to K.L.C. We also thank the Wu Tsai Human Performance Alliance and the Joe and Clara Tsai Foundation for supporting these studies.

## Author contributions

C.J.W. and K.L.C. conceived the study, designed the experiments, and wrote the manuscript. E.G.G., S.C.C., and R.L.S. collected the μCT scans. C.J.W. collected tail tissues and performed H&E staining, EdU pulses, calcein pulses, immunofluorescent imaging, and subsequent image analyses and quantification, as well as RNA extraction for bulk RNA-sequencing experiments. A.J.W. wrote the code used for the μCT analysis pipeline and RNA-seq pipelines. A.J.W., A.Y.L., and C.J.W. carried out the RNA-seq analyses. All authors read and commented on the manuscript.

## Competing interests

The authors declare no competing interests.
