## [Peer Review file · Nature Communications]

Cellular and genetic mechanisms that shape the development and evolution of tail vertebral proportion in mice and jerboas

Corresponding Author: Dr Ceri Weber

Version 0:

Reviewer comments:

Reviewer #1

(Remarks to the Author)

Weber et al. have produced a careful dissection of the processes underlying the unusually long vertebral bodies of the mid-tail vertebrae in jerboas. Their analysis artfully describes tissue, cell, and genetic mechanisms influencing these extreme differences in vertebral proportion. They show that vertebral length differences in the mid-tail—at the height of the bone-length "crescendo"—are the result of changes in vertebral growth rate; this rate difference is driven by increases in the size of the hypertrophic chondrocytes themselves. A valuable component of the paper is the comparison with the authors' work on the evolution of differences in limb bone length. The authors compare gene expression differences among growth plates in mouse and jerboa tail vertebrae and find, intriguingly, that more than 80% of DE genes do not overlap with those found for differences in limb bone length. But among the genes that limb and vertebral elongation do have in common, the authors describe a role for NPR3 in caudal vertebral elongation. Overall, the results in the paper are presented reasonably and claims are not overstated, and the prose is exceptionally clear and well written. The paper's figures are well organized and easy to understand, and the results represent a substantive step forward in the pursuit of the mechanisms responsible for the evolution of skeletal proportion. As such, I really only have minor comments, below:

The authors report that differences in both the number of cells progressing through endochondral ossification and the size of hypertrophic chondrocytes drive differences in vertebral lengths. With the caveat that these are difficult to disentangle, is it possible to estimate the relative contribution of the two mechanisms to differences in vertebral elongation?

The surprising result that the *Npr3*^{-/-} mouse phenotype has longer mid-tail vertebrae, and thus the length difference being the opposite of the direction expected from the jerboa-mouse expression data, is a possible point of confusion/complication, but this result is well considered in the discussion (lines 431–446). It's remarkable that the mid-tail vertebrae, specifically, are longer in these mutants, as they are in naturally occurring variation. The authors note the kyphosis and "cone-shaped implantation" of the tail in the *Npr3*^{-/-} mutants; is there any indication of an axial identity shift in the tail vertebrae mutants?

Line 50: "... forming snakes in the most extreme cases."

Revise for clarity: the process itself doesn't form snakes; it forms, in snakes, the unusually long vertebral column.

Figure 3: I like the colors in the diagrams in A, B, and C and how they are used in the plots in D–F, but panel D and E are not in the order I would expect (i.e., I thought D should be on the left side of the figure and E on the right). Additionally, panel D is not described in the figure caption.

Line 398: "By contrast, we show that hypertrophic chondrocyte cell size differs only the most rapidly elongating jerboa and cells in other vertebrates . . ."

Revise for clarity.

(Remarks on code availability)

I was unable to access the code via the link provided.

Reviewer #2

(Remarks to the Author)

In this manuscript, Weber et al. present an analysis of differential growth in a subset of caudal vertebrae in jerboas, comparing this with the corresponding region in mouse tails. Their analysis uses a combination of approaches, focusing on tail vertebra 6 (TV6) as a representative of the mid-tail region, which exhibits enhanced growth, and comparing it to TV1, representing the proximal tail region. Their findings indicate that the increased size of TV6 in jerboas is not due to increased chondrocyte proliferation. Instead, it results from a greater number of chondrocytes within the proliferative zone and an increase in the height of hypertrophic chondrocytes. To explore the molecular mechanisms underlying this growth pattern, they conduct a comprehensive transcriptomic comparison involving TV1 and TV6 in both jerboas and mice. These analyses revealed a set of genes differentially expressed in jerboa TV6 at the stage when differential growth is most pronounced. Among these, several genes have established roles in regulating skeletal and tail growth, as evidenced by prior mouse gene knockout studies. Among the identified candidates, the authors focus further investigation on *Npr3*, the only gene consistently highlighted across the various differential analyses. They demonstrate that *Npr3*-deficient mice exhibit an enlarged TV1, with its increased size arising from mechanisms similar to those observed in the jerboa's enlarged TV6.

The work presented in this manuscript is technically sound and carefully executed. It provides a detailed anatomical description of the differences in chondrocyte-related features between TV1 and TV6 in both mice and jerboas, with particular emphasis on the pronounced enlargement of TV6 in jerboas. However, the study remains largely descriptive. While it successfully highlights structural and cellular distinctions, it falls short of elucidating the underlying regulatory mechanisms that drive this differential growth. Consequently, it does not fully explain the biological processes responsible for the markedly enlarged vertebrae observed in jerboas.

In my view, understanding the enlarged size of the mid-tail vertebrae in jerboas requires addressing two key aspects:

a) Identification of the genetic and molecular basis for differential vertebral growth. The study highlights a number of genes differentially expressed in TV6, some of which may contribute to the anatomical distinctions observed in jerboas. However, the current analysis does not provide sufficient insight into how these candidate genes might causally influence the pronounced enlargement of the mid-tail vertebrae. In this context, I find the authors' decision to focus on *Npr3* somewhat surprising. While *Npr3* was one of the few genes previously implicated in the regulation of bone size, its known function appears to oppose rather than promote growth. This raises a conceptual challenge in reconciling its elevated expression in jerboa TV6 with a plausible role in driving increased vertebral size.

b) Mechanistic explanation for regional specificity of the phenotype. Another critical question that remains unanswered is why TV6 (or the mid tail vertebrae), specifically, exhibits enhanced growth relative to other regions, here represented by TV1. If this is due to spatially distinct regulation of the genes identified in the transcriptomic analysis, then it becomes essential to clarify how such regional specificity is established. In addition, it is worth noting that global *Npr3* inactivation in mice appears to affect TV1 more significantly than TV6, suggesting that differences in local activity of relevant molecules, possibly involving interactions with other factors, may play a role in region-specific growth regulation. Thus, it would be valuable to explore whether such regional differences arise from variation in gene expression levels alone or from more complex regulatory interactions unique to specific vertebral domains.

(Remarks on code availability)

I tried to access the site but I had a message stating that it is not available

Reviewer #3

(Remarks to the Author)

This paper investigates the mechanisms underlying the differential growth of vertebrae by conducting high-quality cellular and molecular comparisons between jerboa and mouse vertebral growth patterns. Jerboas exhibit disproportionate tail vertebral growth—particularly when comparing TV1 to TV6—whereas mice show relatively consistent growth across the same tail vertebrae. Using detailed histological analyses, the authors suggest that differences in hypertrophic chondrocyte expansion may explain these species-specific growth variations.

Furthermore, they perform interspecies RNA analysis of vertebral cartilage tissues and, through rigorous intersectional analysis, identify several candidate genes associated with enhanced tail vertebral growth in jerboas. Among these, the *Npr3* gene and natriuretic peptide signaling emerge as strong candidates influencing the growth dynamics of jerboa tail vertebrae. The authors then validate this hypothesis by utilizing an *Npr3* knockout mouse model, demonstrating that loss of *Npr3* function results in accelerated cartilage elongation—evidenced by increased chondrocyte hypertrophy and larger hypertrophic cells.

Expression analysis of *NRP3* in jerboas reveals uniform *NRP3* expression in the TV1 growth plate. In contrast, the disproportionately growing TV6 vertebra shows reduced *NRP3* protein levels in the resting zone and pre-hypertrophic chondrocytes, compared to more robust expression in the hypertrophic zone. This suggests that regional variations in *NRP3* protein expression may partly drive the disproportionate growth observed in TV6.

This study provides a compelling mechanistic insight into the role of *NRP3* in regulating disproportionate growth plate dynamics, hypertrophy, and cell size control in chondrocytes. Additionally, it identifies potential factors that may function in

concert with or in parallel to NPR3 in these processes. The authors present strong evidence supporting all their conclusions, and all appropriate controls have been included.

Overall, it was a pleasure to read this manuscript. I recommend it for publication, with only minor revisions needed to improve clarity and readability.

Minor comment on writing style:

Avoid using double adverbs such as “disproportionately differentially” or “significantly differentially,” as they can be confusing. Instead, consider simply stating “differentially expressed genes” with an appropriate threshold. Additionally, it may be helpful to frame the discussion of differentially expressed genes as part of a putative regulatory set that may influence the biological processes under study, rather than making DEGs the primary focus of the sentence.

(Remarks on code availability)

Version 1:

Reviewer comments:

Reviewer #1

(Remarks to the Author)

After reviewing the authors' responses, I'm satisfied that they have addressed my concerns and comments. It was a pleasure to read and review this manuscript and I look forward to seeing it in print.

(Remarks on code availability)

Thanks to the authors for making the code available. It is well annotated and appears to include the information necessary to reproduce the results of the paper.

Reviewer #2

(Remarks to the Author)

The authors have provided a response to my comments on the manuscript; however, their reply relies primarily on general arguments that remain largely speculative at this stage. Notably, there is no substantive effort to explore these points further or to support them with additional analysis or evidence.

I had focused on two main questions that might still need some additional explanation.

1- Identification of the genetic and molecular basis for differential vertebral growth. The authors replied “Identifying the genetic mechanisms of macroevolution over many millions of years is undoubtedly a quantitative genetics (polygenic) problem and is thus unamenable to classical genotype-phenotype approaches”. While I agree with the authors' statement, it is worth noting that they undertook an extensive molecular analysis to uncover the genetic basis of differential vertebral growth. From these analyses, they identified a set of genes associated with skeletal development based on mouse mutant phenotypes. However, aside from Npr3, these candidates were not explored in further detail. If the underlying mechanism indeed involves quantitative genetic contributions, I would anticipate a more comprehensive quantitative analysis that integrates the expression profiles of the differentially expressed genes known to influence skeletal development. Such an approach would better clarify the genetic architecture behind the observed phenotype. Interestingly, Npr3 is essentially the only gene among those identified for which existing genetic data in mice suggests a role in regulating skeletal growth in the opposite direction of what would be expected based on the jerboa phenotype, and therefore, I still find it a strange choice for further analysis even if it is the only common factor coming from the different comparisons, most particularly when assuming a polygenic origin of the phenotype.

The authors also state “However, even if full understanding of the complex regulatory mechanisms of evolution is further from reach, we can identify candidate genes and molecular networks driving relevant biological processes and use these data to understand fundamental mechanisms necessary to establish proportion”. This statement stands in clear contrast to the manuscript's title, which suggests that a mechanistic explanation for the differential morphology of jerboa tail vertebrae is provided. As such, the title may be misleading, as the manuscript does not offer a definitive mechanistic account, but rather a list of potential candidates without in-depth functional validation.

2- Mechanistic explanation for regional specificity of the phenotype. Again, there is no clear explanation for this, just stating that this is an interesting question that one of the members of the group intends to pursue, eventually suggesting the involvement of Hox genes, particularly those from the Hox13 group, mostly based on genetic data from the mouse. If Hox genes are to be taken into account, I would highlight that complete knockout of the Hox11 group results in an even more striking transformation in the identity of the proximal caudal vertebrae, as demonstrated by Wellik and Capecchi (2003, *Science* 301: 363–367). While Hox genes are the typical text book candidates for the regulation of vertebral shape the role of these genes in the differential growth of vertebra TV6 in jerboas remains speculative at this stage, and, without direct experimental assessment does not add much to understand the origin of the differential growth of the middle group of tail vertebra in jerboas. Moreover, other genes have been implicated in modulating tail vertebral morphology and size, such as

Gdf11 (e.g. Aires et al 2019 Dev. Cell 48, 383–395).

I would like to propose a possible developmental mechanism for the tail phenotype of jerboas. During mouse embryogenesis, the tail initially contains a neural tube along its entire length; however, this structure regresses progressively, and by the time of birth (and in adulthood), it is absent beyond the first four or five caudal vertebrae. A similar pattern is observed across most mammalian species. The presence or absence of the neural tube at birth correlates with vertebral shape and size, as evidenced in Hoxb13 and Gdf11 knockout models, likely due to the morphogenetic signals emanating from neural tissues. Within this framework, I propose investigating whether the neural tube extends more caudally in jerboas. Such a difference could offer insight into the developmental mechanisms underlying their distinct vertebral morphology.

(Remarks on code availability)

Reviewer #3

(Remarks to the Author)

The response to reviewers comments are adequate and this paper should be published.

(Remarks on code availability)

NONE

RESPONSE TO REVIEWERS

We appreciate the time and thoughtful consideration given by all three reviewers, especially during these trying times. We are grateful for all suggestions, which have improved the manuscript, and we are appreciative that the reviewers were largely pleased with the writing and presentation of data. A point-by-point response to specific comments can be found below, and we have addressed these in the attached revision.

REVIEWER COMMENTS

Reviewer #1 (Remarks to the Author):

Weber et al. have produced a careful dissection of the processes underlying the unusually long vertebral bodies of the mid-tail vertebrae in jerboas. Their analysis artfully describes tissue, cell, and genetic mechanisms influencing these extreme differences in vertebral proportion. They show that vertebral length differences in the mid-tail—at the height of the bone-length "crescendo"—are the result of changes in vertebral growth rate; this rate difference is driven by increases in the size of the hypertrophic chondrocytes themselves. A valuable component of the paper is the comparison with the authors' work on the evolution of differences in limb bone length. The authors compare gene expression differences among growth plates in mouse and jerboa tail vertebrae and find, intriguingly, that more than 80% of DE genes do not overlap with those found for differences in limb bone length. But among the genes that limb and vertebral elongation do have in common, the authors describe a role for NPR3 in caudal vertebral elongation. Overall, the results in the paper are presented reasonably and claims are not overstated, and the prose is exceptionally clear and well written. The paper's figures are well organized and easy to understand, and the results represent a substantive step forward in the pursuit of the mechanisms responsible for the evolution of skeletal proportion. As such, I really only have minor comments, below:

We are grateful to the reviewer for their kind evaluation of this work.

The authors report that differences in both the number of cells progressing through endochondral ossification and the size of hypertrophic chondrocytes drive differences in vertebral lengths. With the caveat that these are difficult to disentangle, is it possible to estimate the relative contribution of the two mechanisms to differences in vertebral elongation?

This is a great question. Bone elongation is achieved by a combination of proliferation kinetics, matrix synthesis, and chondrocyte enlargement. Wilsman et al (1996) were able to distinguish the relative contributions of these parameters using volumetric reconstructions of limb cartilage captured from >600 images of 1 μm thick plastinated sections¹. Unfortunately, we do not have the capacity to quantify cartilage cellular parameters to this level of resolution.

If we were able to do this, we would additionally collect mean cellular and matrix volumes for the proliferative ($V_{\text{T proliferative zone}}$) and hypertrophic ($V_{\text{T turned over}}$) compartments, numerical densities of these zones ($N_{\text{v proliferative zone}}$; $N_{\text{v terminal hypertrophic zone}}$), and precise total cell-cycle time

per cartilage (Total cell cycle_{hours}). These data, in addition to the elongation rates and proliferation indices we quantified in the present study, would be entered into the equations provided below to calculate the volume of the proliferative zone renewed per day and how much of the hypertrophic zone is lost per day. Although we can't precisely quantify the absolute contribution of the number of cells progressing through endochondral ossification and the size of hypertrophic chondrocytes in this study, and we are grateful to the reviewer for recognizing this caveat in their question, we have compared within and between species to identify *relative* differences between parameters. Further, Wilsman et al suggest that "in steady-state differential growth, parameters of proliferation initiate differential growth, but parameters of hypertrophy amplify these initial differences", which we believe is supported in our data as well¹.

Number of new chondrocytes produced per day:

$$N_{\text{new}} = N_{\text{v proliferative zone}} * V_{\text{T proliferative zone}} * \text{GF (growth fraction)} * (24_{\text{hours}} / \text{Total cell cycle}_{\text{hours}})$$

Number of chondrocytes lost at the chondro-osseous junction per day

$$N_{\text{lost}} = N_{\text{v terminal hypertrophic zone}} * V_{\text{T turned over}}$$

The surprising result that the Npr3^{-/-} mouse phenotype has longer mid-tail vertebrae, and thus the length difference being the opposite of the direction expected from the jerboa-mouse expression data, is a possible point of confusion/complication, but this result is well considered in the discussion (lines 431–446). It's remarkable that the mid-tail vertebrae, specifically, are longer in these mutants, as they are in naturally occurring variation. The authors note the kyphosis and "cone-shaped implantation" of the tail in the Npr3^{-/-} mutants; is there any indication of an axial identity shift in the tail vertebrae mutants?

Indeed, the paradox is confusing, and we are grateful that the reviewer thinks that we addressed it as well as we can at this time. The characteristic kyphosis and "cone-shaped implantation" phenotypes of the NPR3^{-/-} mutants have been noted but not investigated as axial identity shifts in the literature. To address this question, we collected three NPR3^{-/-} mutants and three wildtype siblings at P7 and stained their skeletons with alcian and alizarin dyes. We then counted vertebral elements at each axial level based on morphology. We can confirm that the number of cervical, thoracic, lumbar, sacral, and caudal vertebrae is the same in both mutant and wildtype animals. For this manuscript, we have provided representative images of the sacral and proximal tail skeleton to demonstrate the equivalent morphology of NPR3^{-/-} mutants and wildtype siblings at each vertebral position (see new Supplementary Figure 6).

Line 50: "... forming snakes in the most extreme cases."

Revise for clarity: the process itself doesn't form snakes; it forms, in snakes, the unusually long vertebral column.

Thank you for this suggestion, which is indeed correct :) We have revised this phrasing to "forming the extraordinarily long vertebral column of snakes in extreme cases".

Figure 3: I like the colors in the diagrams in A, B, and C and how they are used in the plots in D–F, but panel D and E are not in the order I would expect (i.e., I thought D should be on the

left side of the figure and E on the right). Additionally, panel D is not described in the figure caption.

Thank you for pointing out that the letters 'D' and 'E' were reversed in Figure 3, and that the descriptions were mislabeled in the figure caption; 'D' was incorrectly referenced as 'E', and 'F' was used twice. We have fixed each of these in the legend.

Line 398: "By contrast, we show that hypertrophic chondrocyte cell size differs only the most rapidly elongating jerboa and cells in other vertebrates . . ."

Revise for clarity.

Thank you for this suggestion. We think that we have improved clarity by revising the last two sentences of this paragraph:

"By contrast, we show that hypertrophic chondrocytes are the same size in mouse vertebrae elongating at different rates and that these are smaller than limb bone hypertrophic chondrocytes. Hypertrophic chondrocytes are only larger in the most rapidly elongating jerboa tail vertebrae suggesting that cell size differs widely in mammal limbs, but perhaps only in vertebrae when a greater difference may contribute to extreme vertebral proportion."

Reviewer #1 (Remarks on code availability):

I was unable to access the code via the link provided.

We apologize that the special link to our Dryad folder giving all Reviewers advance access the data prior to publication was non-functional. Please follow this link to view all RNA-seq data (raw and processed), code, images, and data spreadsheets used in this manuscript.

<http://datadryad.org/share/xvKytHh04JUVWqgPzK2AZawBTHHbGTjBbSM9LT3HQLO>

Reviewer #2 (Remarks to the Author):

In this manuscript, Weber et al. present an analysis of differential growth in a subset of caudal vertebrae in jerboas, comparing this with the corresponding region in mouse tails. Their analyses use a combination of approaches, focusing on tail vertebra 6 (TV6) as a representative of the mid-tail region, which exhibits enhanced growth, and comparing it to TV1, representing the proximal tail region. Their findings indicate that the increased size of TV6 in jerboas is not due to increased chondrocyte proliferation. Instead, it results from a greater number of chondrocytes within the proliferative zone and an increase in the height of hypertrophic chondrocytes. To explore the molecular mechanisms underlying this growth pattern, they conduct a comprehensive transcriptomic comparison involving TV1 and TV6 in both jerboas and mice. These analyses revealed a set of genes differentially expressed in jerboa TV6 at the stage when differential growth is most pronounced. Among these, several genes have established roles in regulating skeletal and tail growth, as evidenced by prior mouse gene knockout studies. Among the identified candidates, the authors focus further investigation on Npr3, the only gene consistently highlighted across the various differential analyses. They demonstrate that Npr3-deficient mice exhibit an enlarged TV1, with its increased size arising from mechanisms similar to those observed in the jerboa's enlarged

TV6.

The work presented in this manuscript is technically sound and carefully executed. It provides a detailed anatomical description of the differences in chondrocyte-related features between TV1 and TV6 in both mice and jerboas, with particular emphasis on the pronounced enlargement of TV6 in jerboas. However, the study remains largely descriptive. While it successfully highlights structural and cellular distinctions, it falls short of elucidating the underlying regulatory mechanisms that drive this differential growth. Consequently, it does not fully explain the biological processes responsible for the markedly enlarged vertebrae observed in jerboas.

We thank the reviewer for their thoughtful synopsis of the work and for their positive evaluation of its technical rigor. We recognize the disappointment with a lack of ‘regulatory mechanism’, but macro-evolutionary mechanism is undoubtedly a quantitative genetics (polygenic) problem and thus difficult to understand by classical genotype-phenotype approaches. We would argue that attribution of ‘mechanism’ has been inappropriately oversimplified in the literature, thus unfortunately setting unrealizable expectations (see a thorough discussion in Cooper, 2024²). That said, we hope the reviewer agrees that we have attributed cellular mechanism and shed light on candidate genes and networks to explain this complex trait. The NPR3 aspect shows potential for such descriptive studies to generate testable hypothesis that can identify fundamental mechanisms of bone growth, even if the complex regulatory mechanisms of evolution are further from reach.

In my view, understanding the enlarged size of the mid-tail vertebrae in jerboas requires addressing two key aspects:

a) Identification of the genetic and molecular basis for differential vertebral growth. The study highlights a number of genes differentially expressed in TV6, some of which may contribute to the anatomical distinctions observed in jerboas. However, the current analysis does not provide sufficient insight into how these candidate genes might causally influence the pronounced enlargement of the mid-tail vertebrae. In this context, I find the authors’ decision to focus on Npr3 somewhat surprising. While Npr3 was one of the few genes previously implicated in the regulation of bone size, its known function appears to oppose rather than promote growth. This raises a conceptual challenge in reconciling its elevated expression in jerboa TV6 with a plausible role in driving increased vertebral size.

We thank the Reviewer for this specific feedback, related to the general comment above, and which readers may also question. To address the subject of ‘mechanism’ in the context of what is likely complex quantitative genetics, we added a brief paragraph in the Discussion:

“Identifying the genetic mechanisms of macroevolution over many millions of years is undoubtedly a quantitative genetics (polygenic) problem and is thus unamenable to classical genotype-phenotype approaches (see Cooper 2024¹⁰⁰). For example, the population diversity of human height, a continuous trait, is associated with thousands of genetic variants that likely each have a very small effect (recently reviewed in Bicknell 2025¹⁰¹); the macroevolution of

skeletal proportion is likely similarly complex. However, even if full understanding of the complex regulatory mechanisms of evolution is further from reach, we can identify candidate genes and molecular networks driving relevant biological processes and use these data to understand fundamental mechanisms necessary to establish proportion.”

Regarding our decision to investigate the function of NPR3, we agree that the paradox is frustratingly confusing. However, we prioritized NPR3 because we found it interesting as the only gene at the intersection of all skeletal proportion datasets, it is one of the few ‘long tail’ mutant mice, and details of the tail phenotype had not been described. We therefore also chose to discuss the paradox, even though we don’t currently understand it.

b) Mechanistic explanation for regional specificity of the phenotype. Another critical question that remains unanswered is why TV6 (or the mid tail vertebrae), specifically, exhibits enhanced growth relative to other regions, here represented by TV1. If this is due to spatially distinct regulation of the genes identified in the transcriptomic analysis, then it becomes essential to clarify how such regional specificity is established. In addition, it is worth noting that global Npr3 inactivation in mice appears to affect TV1 more significantly than TV6, suggesting that differences in local activity of relevant molecules, possibly involving interactions with other factors, may play a role in region-specific growth regulation. Thus, it would be valuable to explore whether such regional differences arise from variation in gene expression levels alone or from more complex regulatory interactions unique to specific vertebral domains.

We thank the reviewer for this stimulating question that we would love to be able to discuss together with them in more depth. We are also fascinated by the regional specificity of the wild type phenotypes and of the NPR loss-of-function phenotype which suggest that vertebrae integrate signaling molecules (e.g. NPR peptides) and cellular contexts differently down the axial skeleton. “How are neighboring vertebrae within a regional domain distinguished from one another?” is a large programmatic question that the lead author, Dr. Weber, is eager to pursue in her independent career.

One possibility is *Hox* gene expression. For example, other groups have shown that *Hoxb13* and *Hoxc13* expression regulate identity within the proximal tail⁴⁻⁶, and *Hoxd13* is associated with tail-length evolution in another rodent species⁷. We wonder if there is a potential role for *Hox* genes to determine individual vertebral size/shape in modulating signal interpretation and have raised this important broader question and specific speculation in the discussion:

“An important remaining question that we are unable to address here is how neighboring vertebrae differentiate in shape and size from one another within a shared regional identity. Because *HoxC13* seems to disproportionately affect the proximal tail, it may play a role in integrating signaling molecules (e.g. NPR peptides) and cellular contexts differently in neighboring tail vertebrae.”

Reviewer #2 (Remarks on code availability):

I tried to access the site but I had a message stating that it is not available

We apologize that the special link to our Dryad folder giving all Reviewers advance access the data prior to publication was non-functional. Please follow this link to view all RNA-seq data (raw and processed), code, images, and data spreadsheets used in this manuscript.
<http://datadryad.org/share/xvKytHh04JUVWqgPzK2AZawBTHHbGTjBbSM9LT3HQLQ>.

Reviewer #3 (Remarks to the Author):

This paper investigates the mechanisms underlying the differential growth of vertebrae by conducting high-quality cellular and molecular comparisons between jerboa and mouse vertebral growth patterns. Jerboas exhibit disproportionate tail vertebral growth—particularly when comparing TV1 to TV6—whereas mice show relatively consistent growth across the same tail vertebrae. Using detailed histological analyses, the authors suggest that differences in hypertrophic chondrocyte expansion may explain these species-specific growth variations.

Furthermore, they perform interspecies RNA analysis of vertebral cartilage tissues and, through rigorous intersectional analysis, identify several candidate genes associated with enhanced tail vertebral growth in jerboas. Among these, the Npr3 gene and natriuretic peptide signaling emerge as strong candidates influencing the growth dynamics of jerboa tail vertebrae. The authors then validate this hypothesis by utilizing an Npr3 knockout mouse model, demonstrating that loss of Npr3 function results in accelerated cartilage elongation—evidenced by increased chondrocyte hypertrophy and larger hypertrophic cells.

Expression analysis of NRP3 in jerboas reveals uniform NPR3 expression in the TV1 growth plate. In contrast, the disproportionately growing TV6 vertebra shows reduced NPR3 protein levels in the resting zone and pre-hypertrophic chondrocytes, compared to more robust expression in the hypertrophic zone. This suggests that regional variations in NPR3 protein expression may partly drive the disproportionate growth observed in TV6.

This study provides a compelling mechanistic insight into the role of NPR3 in regulating disproportionate growth plate dynamics, hypertrophy, and cell size control in chondrocytes. Additionally, it identifies potential factors that may function in concert with or in parallel to NPR3 in these processes. The authors present strong evidence supporting all their conclusions, and all appropriate controls have been included.

Overall, it was a pleasure to read this manuscript. I recommend it for publication, with only minor revisions needed to improve clarity and readability.

We express appreciation to the reviewer for this thoughtful summary, and we are very pleased that they enjoyed reading the manuscript.

Minor comment on writing style:

Avoid using double adverbs such as “disproportionately differentially” or “significantly

differentially,” as they can be confusing. Instead, consider simply stating “differentially expressed genes” with an appropriate threshold. Additionally, it may be helpful to frame the discussion of differentially expressed genes as part of a putative regulatory set that may influence the biological processes under study, rather than making DEGs the primary focus of the sentence.

We thank the Reviewer for this feedback and considered each use of “disproportionately differentially” and “significantly differentially” throughout the manuscript. We recognize that after noting the $p < 0.05$ threshold for significance, we no longer need to say ‘significantly’. However, we chose to keep “disproportionately differentially expressed” as this is an important distinction from all ‘differentially expressed’ genes. We hope the Reviewer finds these revisions adequately simplifying.

We are also grateful for the encouragement to contextualize these ‘candidate genes’ in the framework of larger networks/biological processes. We have re-read the manuscript carefully and inserted ‘in regulatory networks’ or emphasized larger regulatory networks and biological processes where appropriate. In response to feedback from Reviewer 2, we also added a brief paragraph to the discussion that emphasizes the likely genetic complexity of macroevolution of skeletal proportion and that we have found “candidate genes and molecular networks driving relevant biological processes”.

References:

1. Wilsman, N. J., Farnum, C. E., Leiferman, E. M., Fry, M. & Barreto, C. Differential growth by growth plates as a function of multiple parameters of chondrocytic kinetics. *J Orthop Res* **14**, 927–936 (1996).
2. Cooper, K. L. The case against simplistic genetic explanations of evolution. *Development* **151**, dev203077 (2024).
3. Bicknell, L. S., Hirschhorn, J. N. & Savarirayan, R. The genetic basis of human height. *Nat Rev Genet* 1–16 (2025) doi:10.1038/s41576-025-00834-1.
4. Godwin, A. R. & Capecchi, M. R. Hoxc13 mutant mice lack external hair. *Genes Dev* **12**, 11–20 (1998).

5. Tkatchenko, A. V. *et al.* Overexpression of Hoxc13 in differentiating keratinocytes results in downregulation of a novel hair keratin gene cluster and alopecia. *Development* **128**, 1547–1558 (2001).
6. Economides, K. D., Zeltser, L. & Capecchi, M. R. Hoxb13 mutations cause overgrowth of caudal spinal cord and tail vertebrae. *Dev Biol* **256**, 317–330 (2003).
7. Kingsley, E. P. *et al.* Adaptive tail-length evolution in deer mice is associated with differential Hoxd13 expression in early development. *Nat Ecol Evol* **8**, 791–805 (2024).